# EFFECTIVE MODEL SPARSIFICATION BY SCHEDULED GROW-AND-PRUNE METHODS

**Xiaolong Ma[1,†], Minghai Qin[2,†], Fei Sun[2,†], Zejiang Hou[3], Kun Yuan[2], Yi Xu[4],**
**Yanzhi Wang[1], Yen-Kuang Chen[2], Rong Jin[2], Yuan Xie[2]**

[1] Northeastern University [2] DAMO Academy, Alibaba Group [3] Princeton University
[4] Dalian University of Technology

## ABSTRACT

Deep neural networks (DNNs) are effective in solving many real-world problems. Larger DNN models usually exhibit better quality (e.g., accuracy) but their excessive computation results in long inference time. Model sparsification can reduce the computation and memory cost while maintaining model quality. Most existing sparsification algorithms unidirectionally remove weights, while others randomly or greedily explore a small subset of weights in each layer for pruning. The limitations of these algorithms reduce the level of achievable sparsity. In addition, many algorithms still require pre-trained dense models and thus suffer from large memory footprint. In this paper, we propose a novel scheduled grow-and-prune (GaP) methodology without having to pre-train a dense model. It addresses the shortcomings of the previous works by repeatedly growing a subset of layers to dense and then pruning them back to sparse after some training. Experiments show that the models pruned using the proposed methods match or beat the quality of the highly optimized dense models at 80% sparsity on a variety of tasks, such as image classification, objective detection, 3D object part segmentation, and translation. They also outperform other state-of-the-art (SOTA) methods for model sparsification. As an example, a 90% non-uniform sparse ResNet-50 model obtained via GaP achieves 77.9% top-1 accuracy on ImageNet, improving the previous SOTA results by 1.5%. Code available at: `https://github.com/boone891214/GaP`.

## 1 INTRODUCTION

Deep neural networks (DNNs) have achieved great performance in many real-world scenarios. However, the large computation and memory requirements of deep neural networks discourage them from being applied to broader applications on resource-limited devices. Model compression via weight pruning is a popular research topic, where a large proportion of weights are set to zero, leading to significant reduction in both memory and computation. The introduction of sparse tensor cores in the NVIDIA A100 GPU (NVIDIA, 2020b) brings weight pruning into mainstream.

Early works on weight pruning generally follow a **prune-from-dense** methodology (Guo et al., 2016; Yu et al., 2018; Wen et al., 2016; Liu et al., 2019b; Pham et al., 2018; Real et al., 2019), which usually requires 3 phases of training: pre-train a dense model, prune it to sparse, and fine-tune it. In such methodologies, the one-shot or iteratively pruning from a well-trained DNN can only *remove* weights, which lacks the flexibility of growing back weights that are considered unimportant early in the training process but showed to be significant later in training. On the other hand, **early-stage pruning** methods (Lee et al., 2019; Wang et al., 2020; You et al., 2020) avoid training dense models to converge. However, those sparse masks (i.e., the binary matrix in which a zero value removes the corresponding weight entry from the model) are prematurely fixed, resulting in inferior model quality. Methods based on **sparse mask exploration**, such as DeepR (Bellec et al., 2018), SET (Mocanu et al., 2018), DSR (Mostafa & Wang, 2019), and RigL (Evci et al., 2020) maintain the target sparsity in all layers throughout the training process and selectively explore a small fraction of the weights

---

† Equal Contribution.
This work is partially done during Xiaolong Ma's internship and Yi Xu's working period at Alibaba Group.

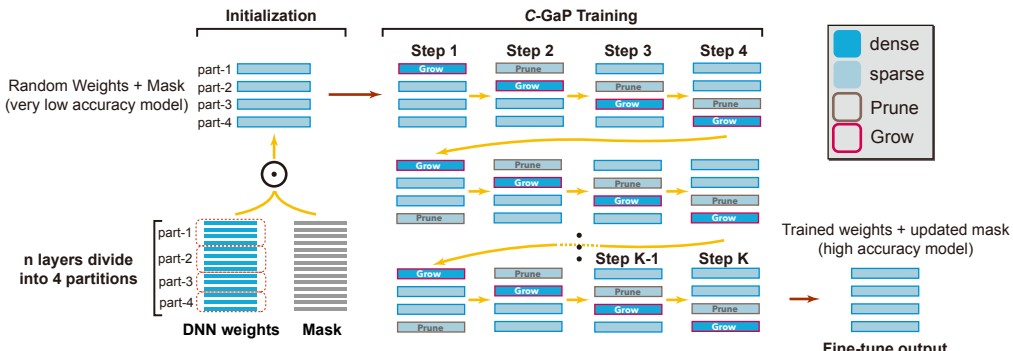

Figure 1: Overview of the cyclic GaP (*C*-GaP) training method. We assume 4 partitions in a model are grown and pruned in a cyclic order. Only one partition is kept dense during training. After $K$ steps, the dense partition is pruned and the whole model is fine-tuned to obtain the sparse model.

periodically. The explored weights are chosen based on random or greedy heuristics, leading to limited coverage of sparse patterns and consequentially sub-optimal model quality.

In order to overcome the shortcomings of the previous solutions, we propose a scheduled grow-and-prune (GaP) methodology. It does not require a dense model any time during the training process, leading to largely-reduced memory footprints. Meanwhile, the sparse mask of a layer is updated after exploring all weights in the same layer, resulting in better mask-update efficiency. It targets to obtain the best quality sparse model under a runtime budget. Since some large models may be compressed below a target latency before being deployed in the wild to thousands of devices and inferenced billions of times per day (Hazelwood et al., 2018; Wu et al., 2019), spending a bit more time in training to find a better model is well rewarded.

The scheduled GaP methodology divides a DNN into several partitions composed of contiguous layers. During model training, one of the sparse partitions is grown to dense while the rest remain sparse. After some epochs of training, the previously dense partition is pruned to sparse, and an alternate partition is grown to dense. This process is repeated so that all partitions are grown to dense and pruned back to sparse multiple times. If the partitions are grown to dense one by one sequentially, it is called the *cyclic* GaP method (as shown in Figure 1). If the model is replicated to multiple machines with different dense partitions, it is called the *parallel* GaP method.

The GaP methods are carefully scheduled to explore all weights efficiently, which is not guaranteed in the existing mask exploration methods. We illustrate their differences by an example in Figure 2. In the scheduled GaP methods, all weights to be grown (or to be pruned) belong to the same partition. All weights in the model are explored when every partition is grown to dense and pruned to sparse. On the other hand, in the random or greedy exploration methods, the grown and pruned weights are distributed across all layers and they do not guarantee that all weights are explored, e.g, the connection between 1st input neuron and 3rd neuron in the first layer is not explored in Figure 2(b). Random sparse mask exploration and our scheduled GaP methods are like sampling operations with replacement and without replacement (Basu, 1958). The former requires a lot more samples to achieve the same weight coverage as the latter. Please see Appendix C for an illustrative example.

The contribution of this paper is summarized as follows.

- We propose two GaP methods such that they do not require to train dense models any time throughout the training process, reducing the training memory footprints.

- Sparse models obtained by the proposed GaP methods outperform the SOTA sparsification algorithms on a wide variety of deep learning tasks, including 2D image recognition, object detection, 3D object part segmentation, and machine translation.

- For all four tasks, the proposed GaP methods surprisingly generate sparse models that match or even exceed the dense counterparts at 80% sparsity.

- We provide theoretical guarantees for the GaP method and clarify its convergence property. In particular, we justify the influence of the mask-induced error.

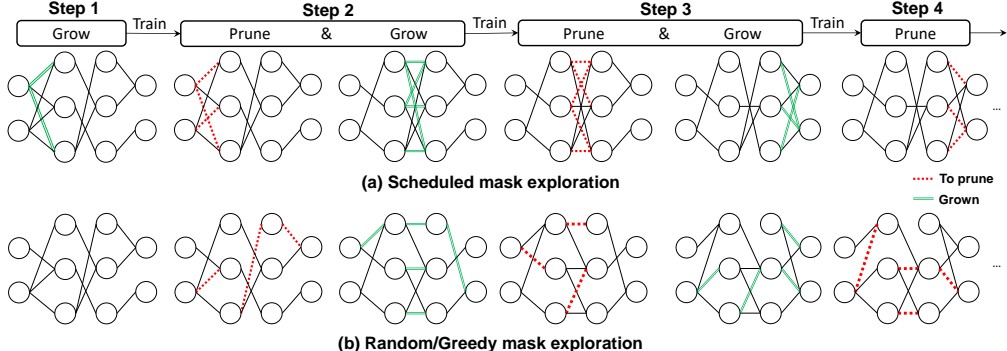

Figure 2: Comparison between (a) the scheduled mask exploration and (b) the random/greedy mask exploration. In (a), the weights to be grown (or pruned) are in the same layer and the growing phase covers all weights in that layer. In (b), they may be in different layers and the growing phase does not guarantee a coverage during training.

---

**Algorithm 1:** $C$-GaP training flow.

---

**Input:** An $L$-layer model with uninitialized weight $\Theta$; pruning ratio $r$.
**Output:** An $L$-layer sparse model satisfying the target sparsity requirement.
1 Initialize $f(x; \Theta \odot m)$ with random weights $\Theta$ and random masks $m$ that satisfy the sparsity requirement.
2 Divide all layers into $\kappa$ partitions, denoted by $S_i, i \in \{0, 1, \ldots, \kappa - 1\}$.
3 $step = 0$
4 **while** $step < K$ **do**
5      Partition to grow index $i \leftarrow step \bmod \kappa$, partition to prune index $j \leftarrow (step - 1) \bmod \kappa$.
6      Prune $\Theta^{S_j}$ and update $m^{S_j}$ by $[\Theta^{S_j}, m^{S_j}] \leftarrow \texttt{ArgPruneTo}(\Theta^{S_j}, r)$.
7      Grow $\Theta^{S_i}$ by $m^{S_i} \leftarrow \texttt{ArgGrowTo}(\Theta^{S_i})$.
8      Train the model $f(x; \Theta \odot m)$ for $T$ epochs.
9      $step = step + 1$.
10 Denote the final dense partition by $S_d$.
11 Prune $\Theta^{S_d}$ and update $m^{S_d}$ by $[\Theta^{S_d}, m^{S_d}] \leftarrow \texttt{ArgPruneTo}(\Theta^{S_d}, r)$.
12 Fine-tune $f(x; \Theta \odot m)$ for $T'$ epochs.

---

## 2 METHODOLOGY

In this section, we describe the scheduled grow-and-prune (GaP) methods in detail. The process starts from a randomly initialized model. First, its weights are pruned randomly to reach the target sparsity, i.e. a random sparse mask is applied to the weights. Then, the model is divided into $\kappa$ partitions and each partition is grown and pruned separately. In the following two sub-sections, two variants of the GaP methods are described in detail.

### 2.1 CYCLIC GAP

As shown in Figure 1, the cyclic GaP ($C$-GaP) method rotates the dense partitions among all $\kappa$ partitions ($\kappa = 4$ in Figure 1). Starting from all partitions with random sparse masks, the first partition is grown to a dense one. All weights in the first partition and the masked weights in the remaining 3 partitions are trained for a few epochs (Step 1). Then, the first partition is pruned to sparse (Step 2). Next, we apply the same strategy to the second partition, then continue iterating over all $\kappa$ partitions. When all layers are cyclically grown and pruned once after $\kappa$ steps, we call it *one round*. This process is repeated $K$ times.

Algorithm 1 describes this process in more detail. In Line 1, we use $f(x; \Theta)$ to represent an $L$-layer deep neural network model, where $\Theta \in \mathbb{R}^M$ are $M$ trainable weights and $x$ are training samples. A sparse model is denoted as $f(x; \Theta \odot m)$, where $\odot$ denotes element-wise multiplication. $m$ is a binary mask $m \in \{0, 1\}^M$, where a zero in $m$ means that the corresponding weight in $\Theta$ is fixed to be zero and a one in $m$ means that the corresponding weight in $\Theta$ is free to be updated.

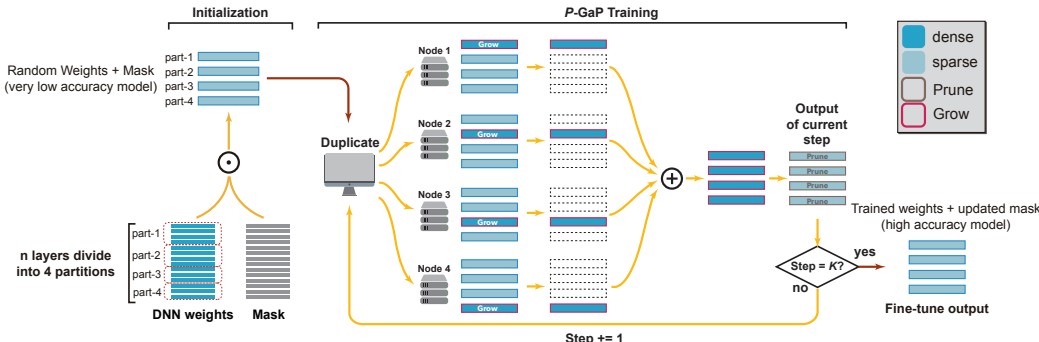

Figure 3: Overview of the parallel GaP (*P*-GaP) training method. We assume 4 training nodes are used for the 4 partitions, which are grown in parallel for $T$ epochs. At the end of each step, the dense parts in the nodes are combined, followed by a pruning back to a sparse model. After $K$ steps, the model is fine-tuned.

In the $C$-GaP, all layers of the model $\{1, 2, \ldots, L\}$ are divided into $\kappa$ subsets $S_0, S_1, \ldots, S_{\kappa-1}$, where $\kappa$ is an integer between 1 and $L$, as shown in Line 2. Let $\Theta^S$ and $m^S$ denote the subset of trainable weights and their masks. Consequently, $\Theta$ is divided into $\kappa$ partitions, denoted by $\Theta^{S_0}, \ldots, \Theta^{S_{\kappa-1}}$. For the $i$-th partition $\Theta^{S_i}$, we denote its mask by $m^{S_i}$.

Then the grow and prune process is iterated $K$ times (Line 4). First, the indices of the partitions to grow and prune are calculated (Line 5). They are determined in a cyclic order. Then, the selected dense partition is pruned to sparse (Line 6). It prunes the partition $\Theta^{S_j}$ with a pruning ratio $r$. In the mean time, the next sparse partition is grown to dense (Line 7). It sets the mask $m^{S_i}$ to be all ones such that all weights $\Theta^{S_i}$ are free to be updated during training. After that, the model is trained for $T$ epochs (Line 8). After the entire grow-and-prune process is finished, the final dense partition is pruned to sparse (Line 11) and the model is fine-tuned to get the final trained sparse model (Line 12).

## 2.2 PARALLEL GaP

In this section, we introduce the parallel GaP (*P*-GaP) as a flexible and highly efficient complement to the *C*-GaP sparse training method when sufficient training resources are available, as described in Figure 3. Unlike the *C*-GaP method where partitions are grown and pruned serially in one machine, the *P*-GaP method distributes the model to $\kappa$ machines. Each grows a different partition to dense and trains the model in parallel. Then, the trained dense partitions are combined to a dense model and pruned to a sparse one again. As a result, the *P*-GaP method utilizes $\kappa$ more machines, but its training time is also $\kappa$ times shorter than the *C*-GaP method. Since each server trains the model independently, the inter-server communication only occurs when the dense parts are combined to synchronize their learned weights once every few epochs. Thus its impact to performance is negligible. Each pair of distribution and combining form one round in the *P*-GaP. Unlike *C*-GaP, one round of *P*-GaP consists of only one step. The detailed *P*-GaP method is shown in Algorithm 2 and illustrated in Figure 4 (b) in Appendix D.

Some of the differences between the parallel GaP and the conventional distributed training are listed as follows.

1. In *P*-GaP, data communicates across different server nodes every $T$ epochs of training. It is far less frequent than conventional distributed training, which requires communications and synchronization in every mini-batch. Therefore, the data bandwidth between the nodes remains minimal.

2. In *P*-GaP, each node can use the single-node hyper-parameters to train the sparse models. It does not require excessively large batch-size to fully utilize the computing resource.

3. Different from data parallelism, model parallelism, and pipeline parallelism, the *P*-GaP explores another dimension of parallelism when training sparse models, which we call it *mask parallelism*. This implies that masks in different partitions are less correlated and can be searched separately.

Table 1: Results of sparse ResNet-50 models on ImageNet.

| Method | Distribution | Total epochs | Sparsity ratio: 80% | | Sparsity ratio: 90% | |
|---|---|---|---|---|---|---|
| | | | Top-1 Acc (%) | FLOPS▲ (×e9) | Top-1 Acc (%) | FLOPS▲ (×e9) |
| Dense (NVIDIA, 2020a) | | Top-1 accuracy: 78.2, FLOPS: 8.2×e9 | | | | |
| Prune-from-dense | uniform | 750/1250† | 77.1 | 1.7 | 75.8 | 0.8 |
| SNIP (Lee et al., 2019) | uniform | 100 | 72.0 | 1.7 | 67.2 | 0.8 |
| SET (Mocanu et al., 2018) | uniform | 100 | 72.9 | 1.7 | 69.6 | 0.8 |
| $SET_{12\times}$* | uniform | 1200 | 76.4 | 1.7 | 74.5 | 0.8 |
| RigL (Evci et al., 2020) | uniform | 100 | 74.6 | 1.7 | 72.0 | 0.8 |
| RigL* | uniform | 100 | 74.6 | 1.7 | 72.5 | 0.8 |
| $RigL_{5\times}$ (Evci et al., 2020) | uniform | 500 | 76.6 | 1.7 | 75.7 | 0.8 |
| $RigL_{5\times}$* | uniform | 500 | 76.9 | 1.7 | 75.6 | 0.8 |
| $RigL_{12\times}$* | uniform | 1200 | 77.1 | 1.7 | 76.0 | 0.8 |
| *C*-GaP | uniform | 990 | **77.9** | 1.7 | **76.3** | 0.8 |
| *P*-GaP | uniform | 1110 | 77.5 | 1.7 | 76.1 | 0.8 |
| GMP (Gale et al., 2019) | non-uniform | 135 | 76.5 | n/a | 75.0 | n/a |
| SNIP (Lee et al., 2019) | non-uniform | 100 | 69.7 | 2.8 | 61.9 | 1.9 |
| GraSP (Wang et al., 2020) | non-uniform | 150 | 72.1 | 2.8 | 68.1 | 1.9 |
| DeepR (Bellec et al., 2018) | non-uniform | 100 | 71.7 | n/a | 70.2 | n/a |
| SET (Mocanu et al., 2018) | non-uniform | 100 | 72.6 | 1.7 | 70.4 | 0.8 |
| DSR (Mostafa & Wang, 2019) | non-uniform | 100 | 73.3 | 2.9 | 71.6 | 0.8 |
| DPF (Lin et al., 2020b) | non-uniform | 90 | 75.1 | n/a | n/a | n/a |
| RigL (Evci et al., 2020) | ERK | 100 | 75.1 | 3.4 | 73.0 | 2.0 |
| RigL* | ERK | 100 | 75.4 | 3.4 | 73.9 | 2.0 |
| $RigL_{5\times}$ (Evci et al., 2020) | ERK | 500 | 77.1 | 3.4 | 76.4 | 2.0 |
| $RigL_{5\times}$* | ERK | 500 | 77.4 | 3.4 | 76.3 | 2.0 |
| $RigL_{12\times}$* | ERK | 1200 | 77.4 | 3.4 | 76.8 | 2.0 |
| *C*-GaP | non-uniform | 990 | **78.1** | 2.7 | **77.9** | 2.0 |

† 750 epochs for 80% sparsity and 1250 epochs for 90% sparsity (please see Appendix B.1 for details).
* Our implementation (please refer to Appendix B.2 for implementation details).
▲ Following the convention in (Evci et al., 2020), multiplication and addition are counted as two operations.

# 3 EXPERIMENTAL RESULTS

This section evaluates the proposed *C*-GaP and *P*-GaP sparse training methods on different machine learning tasks. Since our goal is to search for the optimal sparse models to speed up inference, we report the best model accuracy obtained using the GaP methods for all models. We compare our GaP method with the state-of-the-arts, as well as our implemented prune-from-dense results. If not otherwise specified, all prune-from-dense are implemented using magnitude-based iterative pruning schedule as described in GMP (Zhu & Gupta, 2017; Gale et al., 2019). More ablation experiments on partition number, partition scheme, different sparsity granularity using GaP, and FC layer pruning are shown in Appendix A.

All of our experimental results are trained and inferenced using PyTorch in the machines with 8 NVIDIA-V100 GPUs. The cyclic GaP is trained in one training node and the parallel GaP is trained in $\kappa$ training nodes, where $\kappa$ is the number of model partitions. In the pruning stage of the GaP method, the weights with the lowest magnitudes are pruned to zero. Note that in our experiments, only the weights for convolution, matrix-vector, and matrix-matrix multiplications are made sparse. The weights related to biases and batch-normalization are kept dense, as they are critical for model quality but only contribute to a tiny fraction of the inference FLOPs. For uniform sparsity, we prune individual layer separately by sorting all weights in the layer based on their magnitudes, and prune away any weight whose magnitude is below the percentile described as the sparsity level. Thus, all layers receive the same sparsity. For non-uniform sparsity, the pruning process is similar, but the weights in the entire model are sorted together. Thus, the sparsity level in different layers may differ, with the more important layers pruned less than the trivial layers. We list the implementation details and hyper-parameter settings for all tasks in Appendix B.

## 3.1 Image classification: ResNet-50 on ImageNet

Table 1 compares the accuracy between using the GaP methods and the previous works. The ResNet-50 is divided into four partitions by the last three downsampling stages. For non-uniform sparsity, all layers are sparsified. For the uniform sparsity, the first convolutional layer with $7 \times 7$ kernels is kept dense, the same as in Evci et al. (2020). The fully connected (FC) layer only contributes 0.5% of FLOPs, thus it is also kept dense (sparse FC layer has similar accuracy with dense FC layer, please check the ablation results in Appendix A).

To ensure fair comparison, we perform the following demonstrations in addition to the original baseline results. We include the SNIP (Lee et al., 2019) and SET (Mocanu et al., 2018) results in uniform sparsity that are reproduced in Evci et al. (2020). We also implement the RigL and $\text{RigL}_{5\times}$ (Evci et al., 2020) using our code base on PyTorch (please refer to Appendix B.2 for details). In Table 1, our reproduction achieves equivalent or higher accuracy than the one reported in the paper. Based on that, we extend the RigL training time by $12\times$ to match the GaP method. Additionally, SET shares similar method with RigL, thus we also implement and extend SET training by $12\times$.

We observe that the improvement over $\text{RigL}_{5\times}$ (Evci et al., 2020) is 1.3% (77.9% vs. 76.6%) at uniform 80% sparsity, and 0.6% at 90% uniform sparsity. The $C$-GaP is also better than our re-implementation of the prune-from-dense method using ADMM (Ren et al., 2019). For non-uniform sparse ResNet-50 model, the improvement over $\text{RigL}_{5\times}$ (Evci et al., 2020) is 1.0% and 1.5% at 80% and 90% sparsity, respectively. Note that the 80% sparse ResNet-50 model can almost match the accuracy of the dense model reported by NVIDIA (2020a). We also observe that, when we extend the training time for the SOTA methods such as SET and RigL to match the GaP training efforts, their accuracies still lag behind. As an example, the model at 80% non-uniform sparsity trained using the $C$-GaP method outperforms the re-implementation of the ERK $\text{RigL}_{12\times}$ model with 0.7% accuracy increase and 21% FLOPS reduction (2.7 GFLOPS vs. 3.4 GFLOPS). The 90% non-uniform sparsity model achieves 1.1% accuracy increase at the same FLOPS.

Comparing the $C$-GaP and $P$-GaP pruning methods with same training epochs, we find that $C$-GaP achieves 0.4% and 0.2% higher accuracy at 80% and 90% uniform sparsity, respectively. We conjecture the reason is that $P$-GaP converges slower than $C$-GaP. In $C$-GaP, the weight values in the sparse partitions are continuously trained to get better model accuracy when the dense partition is explored, while the $P$-GaP method only keeps the dense partition and discards the sparse partitions when combining all dense partitions to the updated model.

## 3.2 Object detection: SSD on COCO-2017

We divide the SSD network into 4 partitions with three in the backbone and one in the detection head. We train the model using cyclic-partitioned $C$-GaP with a uniform weight sparsity of 90% on all layers except for the last output layer, which is kept dense. We report the accuracy of the best model searched within 40 GaP steps. As shown in Table 2, in the mean average precision category mAP@[.5:.95], the sparse model obtained using our $C$-GaP method exceeds the dense model by 0.7 mAP (25.9 vs 25.2). It also exceeds the best sparse model iteratively pruned from the dense model by 1.6 mAP (25.9 vs 24.3).

Table 2: Results of sparse SSD models for object detection on COCO-2017.

| Method | Sparsity | AP, IoU: | | | AP, Area: | | | AR, #Dets: | | | AR, Area: | | |
|---|---|---|---|---|---|---|---|---|---|---|---|---|---|
| | ratio | 0.5:0.95 | 0.5 | 0.75 | S | M | L | 1 | 10 | 100 | S | M | L |
| Dense (NVIDIA, 2020a) | 0 | 25.2 | 42.7 | 25.8 | 7.3 | 27.1 | 40.8 | 23.8 | 34.5 | 36.1 | 11.7 | 39.6 | 56.1 |
| Prune-from-dense | 90% | 24.3 | 41.1 | 24.8 | 6.8 | 26.3 | 40.0 | 23.3 | 34.0 | 35.8 | 11.1 | 39.4 | 55.7 |
| **$C$-GaP** | 90% | **25.9** | 42.3 | 26.9 | 8.0 | 28.1 | 42.6 | 24.7 | 35.9 | 37.8 | 12.7 | 41.5 | 58.5 |

## 3.3 3D object part segmentation: PointNet++ on ShapeNet

We divide the PointNet++ model into 4 partitions with three in the backbone and one in the segmentation head. We apply the $C$-GaP and $P$-GaP methods with uniform sparsity of 80% and 90% on all layers, respectively. We report the accuracy of the best model searched within 40 GaP steps. Table 3 compares them with the dense model and the best sparse model pruned from the dense model using

the ADMM algorithm. The results show that on the class and instance mAP categories, the pruned model using the *C*-GaP method easily beats the model pruned from dense at both sparsities. It even beats the dense model at 80% sparsity. The pruned model using the *P*-GaP method also beats the model pruned from dense at 80% sparsity and is not far behind at 90% sparsity.

Table 3: Results of sparse PointNet++ models for 3D part segmentation on ShapeNet.

| Method | #Points | Sparsity ratio: 80% | | Sparsity ratio: 90% | |
| --- | --- | --- | --- | --- | --- |
| | | Class mIoU (%) | Instance mIoU (%) | Class mIoU (%) | Instance mIoU (%) |
| Dense (Yan, 2019) | 2k | Class mIoU: 82.5, Instance mIoU: 85.4 | | | |
| Prune-from-dense | 2k | 79.1 | 84.5 | 77.1 | 84.0 |
| *C*-GaP | 2k | **82.9** | **85.8** | **79.5** | **85.1** |
| *P*-GaP | 2k | 80.8 | 85.5 | 74.0 | 83.7 |

## 3.4 TRANSLATION: TRANSFORMER ON WMT-14 EN-DE DATASET

In this part, we evaluate the *C*-GaP and the *P*-GaP methods on the translation task based on Transformer (Vaswani et al., 2017). We train the Transformer models on the WMT-14 En-De dataset and evaluate the SacreBLEU score (Post, 2018).

We equally divide the Transformer models into three or six partitions with the decoder containing $2\times$ partitions of the encoder. We apply the *C*-GaP and *P*-GaP methods with uniform sparsity of 80% and 90% on all layers, respectively.

We report the best SacreBLEU scores on the validation dataset in Table 4 within 30 GaP steps. The models trained using both *C*-GaP and *P*-GaP methods significantly improve over the sparse models obtained by pruning from the dense counterparts. They exceed the dense model quality at 80% sparsity, and the model using three-partition *C*-GaP method even outperforms the dense model at 90% sparsity.

Table 4: Results of sparse Transformer models for the translation task on WMT14 En-De.

| Method | Sparsity ratio: 80% | | Sparsity ratio: 90% | |
| --- | --- | --- | --- | --- |
| | $\kappa = 3$ | $\kappa = 6$ | $\kappa = 3$ | $\kappa = 6$ |
| Dense (NVIDIA, 2020a) | SacreBLEU: 27.6 | | | |
| Prune-from-dense | 27.1 | | 25.7 | |
| *C*-GaP | 27.6 | 27.6 | **27.7** | 27.1 |
| *P*-GaP | **27.9** | 27.7 | 27.3 | 26.9 |

## 4 THEORETICAL JUSTIFICATION

We now provide the convergence guarantee for the cyclic-partitioned *C*-GaP algorithm. We let $F(\Theta) = \mathbb{E}_{x \sim D} f(x; \Theta)$ be the loss function of the deep learning task where $D$ is the data distribution. In addition, we let $m_q$ be the mask at the start of the $q$-th round, $\Theta_q$ be the learned model weights after $q-1$ rounds, and $\Theta_q^{(i)}, i = 1, \cdots, \kappa$ be the learned weights in the $q$-th round after the $i$-th partition is grown and trained for $T$ epochs. Proposition 1 shows that *C*-GaP converges to a neighborhood around the stationary solution at rate $O(1/\sqrt{Q})$ when learning rate is set properly. Due to the space limitation, we put its proof in Appendix E.

**Proposition 1** (*C*-GAP CONVERGENCE). *Suppose the loss function $F(\Theta)$ is partition-wise L-smooth, the sampled stochastic gradient is unbiased and has bounded variance, and the relative error introduced by each mask is bounded by $\delta^2 \in [0, 1)$, i.e., $\|\Theta_q - \Theta_q \odot m_q\|^2 \leq \delta^2 \|\Theta_q\|^2$. If the learning rate $\gamma = 1/(4\kappa LT\sqrt{Q})$, the sparse models generated by the C-GaP method after Q rounds will converge as follows:*

$$\frac{1}{Q} \sum_{q=1}^{Q} \mathbb{E}\|\nabla F(\Theta_q \odot m_q)\|^2 = O\Big(\frac{G}{\sqrt{Q}} + \frac{\delta^2}{Q} \sum_{q=1}^{Q} \sum_{i=1}^{\kappa} \mathbb{E}\|\Theta_q^{(i)}\|^2\Big) \tag{1}$$

*where G is a constant related to the gradient noise and the initialized model weights.*

We make several remarks for Proposition 1 as follows.

**Remark 1.** *If there is no pruning in the training process, then it holds that $\delta^2 = 0$. Substituting it into equation 1, we find that C-GaP will converge exactly to a stationary solution, i.e., $\mathbb{E}\|\nabla F(\Theta_Q)\|^2 \to 0$ as Q increases to infinity. This implies C-GaP is also an effective algorithm even for dense training.*

**Remark 2.** *When a sparse mask is utilized, it holds that $\delta^2 \neq 0$. In this scenario, the mask-induced error will inevitably influence the accuracy of the trained model. A smaller mask-induced error will lead to a more accurate model that C-GaP can converge to.*

**Remark 3.** *In our proofs, we find that cycling through all partitions so that all weights can be trained per round is critical to guarantee the convergence of C-GaP. Note that previous works update weights either greedily (e.g., RigL (Evci et al., 2020)) or randomly (e.g., SET (Mocanu et al., 2018) and DSR (Mostafa & Wang, 2019)). It may take numerous training steps for these algorithms to have each weight explored and updated. This may explain why C-GaP achieves better accuracy than RigL, SET, and DSR (see Table 1). This intuition is consistent with (Gürbüzbalaban et al., 2019; Ying et al., 2018) which theoretically establish that sampling data cyclically converges faster than uniformly randomly in SGD training.*

## 5 RELATED WORKS

**Pruning from pre-trained dense models**: Among various pruning algorithms, one-shot pruning (Le-Cun et al., 1990; Xu et al., 2021) zeros out a given percentage of the trained weights with the smallest magnitude. Based on the magnitude-based one-shot pruning, the gradual magnitude pruning (GMP) (Zhu & Gupta, 2017; Gale et al., 2019) proposes an iterative pruning method that progressively prune models to the target pruning ratio. However, those methods suffer from significant accuracy drop and relatively long training time. Besides magnitude-based pruning, other approaches (Wen et al., 2016; He et al., 2017; Zhang et al., 2021a) adapt mathematics-oriented regularization-based algorithms to generate sparsity. Later works (Zhang et al., 2018b; Ren et al., 2019; Ma et al., 2020a;b; Niu et al., 2020; 2021; Rumi et al., 2020; Ma et al., 2021a; Zhang et al., 2021b; Huang et al., 2021) utilizes dynamic regularization penalty such as ADMM to solve the pruning problem as an optimization problem and maintaining high accuracy. Several methods (Zhou et al., 2021; Srinivas et al., 2017; Molchanov et al., 2017) work in probability space. Other works such as HRank (Lin et al., 2020a), SCOP (Tang et al., 2020), DMCP (Guo et al., 2020), MetaPruning (Liu et al., 2019a) use complicated rules to generate the sparsity distribution in the channel level. Since pre-training a dense model from which to select critical weights consumes large memory space, our work differs substantially from them such that we do not rely on a pre-trained dense model.

**Pruning at an early stage**: A new trend of exploring sparsity at an early stage (Lee et al., 2019; Wang et al., 2020; Tanaka et al., 2020; Wimmer et al., 2020; van Amersfoort et al., 2020; Ma et al., 2021b; Liu et al., 2021) has emerged to embrace the promising sparse training paradigm. SNIP (Lee et al., 2019) finds the sparse masks based on the saliency score of each weight that is obtained after training the dense model for only a few iterations. GraSP (Wang et al., 2020) prunes weights based on preserving the gradient flow in the model. EarlyBird (You et al., 2020) conducts a retraining process after searching sub-network topology for a few epochs. PruneTrain (Lym et al., 2019) integrates structured pruning in the pretraining process. However, pruning at an early stage fails to achieve acceptable accuracy on ImageNet (Deng et al., 2009). We recognize the underlying reason is that the single-shot pruning criterion is based on the information from a few mini-batch of training samples, which cannot accurately represent the rich features of large-scale dataset. The proposed GaP methods repeatedly update the sparse masks and hence overcome their drawbacks.

**Sparse mask exploration**: To satisfy the condition of low computation cost as well as low memory footprint, many works have incorporated the training-from-sparse-initialization paradigm, and optimize the network by sparse mask exploration. DeepR (Bellec et al., 2018) proposes to train a sparse network at initialization, and dynamically adjust the sparse topology by pruning the weights that flip the sign and randomly growing new weights during training. However, DeepR is primarily demonstrated with sparsification of fully-connected layers and applied to relatively small and shallow networks. Sparse Evolutionary Training (SET) (Mocanu et al., 2018) and MEST (Yuan et al., 2021a) uses layer-wise magnitude-based pruning and random growth at the end of each training epoch.

DSR (Mostafa & Wang, 2019) and STR (Kusupati et al., 2020) designs a dynamic reparameterization method that allows weights to be re-distributed across layers by providing an effective and trainable way to allocate the sparsity across all layers. Similarly, DNW (Wortsman et al., 2019) trains a sparse model and keeps dense gradients to dynamically adjust the network connections. SNFS (Dettmers & Zettlemoyer, 2019) develops sparse momentum to identify the importance of weights. RigL (Evci et al., 2020) and NeST (Dai et al., 2019) propose to use magnitude-based pruning and gradient-flow-based growth that update sparse model topology during training. Yuan et al. (2021b) uses a continuous relaxation and optimization method to dynamically update the structural masks of DNN models. All of the mentioned works update the sparse masks based on heuristics and there is little or no training at the mask update stage. Our work differs from them in that the mask updating rule of the GaP methods is more optimized by training a subset of layers into dense and then pruned.

## 6 SOCIETAL IMPACT AND LIMITATIONS

**Societal Impact.** The scheduled GaP methods target to obtain the best quality of the pruned model under a runtime budget. Based on the researches in Facebook (Hazelwood et al., 2018; Wu et al., 2019), many models are inferenced many billions of times per day, which makes the cost of pruning such models a tiny fraction of the inference cost in one day. Thus, a better pruning algorithm resulting to a smaller model may save a lot more computation on the inference side, which is of high value for the community and society to achieve Green AI (Schwartz et al., 2020). On the other hand, our method cannot prevent the possible malicious usage, which may cause negative societal impact.

**Limitations.** In this work, we focus on the unstructured sparsity to demonstrate the algorithm-level innovations of our proposed GaP methods, and we only preliminarily include a block sparsity example in Table 9 of Appendix A.3 to demonstrate its applicability to structured sparsity and its potential to improve the inference speed. More detailed analysis is still a work in progress.

The scheduled GaP methods require more training time than conventional pruning methodology, which may potentially limit its applicability when the training compute resources are limited. Thus, the scheduled GaP methods are mostly beneficial to models whose benefit of the reduced inference time and/or improved accuracy out weigh the cost of the moderately longer training time.

The key hyper-parameter settings such as partition numbers or partition strategies are determined heuristically. In this work, we try to divide the models to partitions with similar parameter counts or computation. We also intuitively group consecutive layers to the same partition, based on our hypothesis that adjacent layers are tighter correlated than more distant layers.

Our analysis in Proposition 1 cannot be directly extended to cover $P$-GaP since the analysis is built on the partition-wise cyclic updating structure of $C$-GaP. We will leave the analysis for $P$-GaP as a future work.

The prediction results from the sparse models are generally biased comparing with the dense models they are pruned from (Hooker et al., 2019). This is a general problem on sparse model architectures, and our GaP methodology cannot reduce such biases.

## 7 CONCLUSIONS

Sparsity based model compression is gaining traction in research and industry to reduce the memory footprint and inference runtime. However, conventional compression approaches result to noticeable accuracy loss during the pruning process. In this paper, we propose a scheduled grow-and-prune (GaP) methodology to obtain sparse models. It divides a model into several partitions, grows and prunes each partition cyclically or in parallel. The experimental results show that this method preserves the model quality far better than previous works. In all four experimented tasks, the GaP method matches or even beats the dense solution at 80% sparsity. In addition, the GaP method does not require a pre-trained dense model and can continuously searching sparsity masks with new data, showing its flexibility and practical potential.

## 8 ACKNOWLEDGEMENT

This work was supported by Alibaba Group through Alibaba Research Intern Program, and was partly supported by the National Science Foundation CCF-1937500. Any opinions, findings, and conclusions or recommendations expressed in this material are those of the authors and do not necessarily reflect the views of NSF.

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

# A  ABLATION STUDY

In this section, we present some ablation study on the proposed GaP methods.

## A.1  NUMBER OF PARTITIONS FOR GAP

The number of partitions in GaP methods play a key role in obtaining better results. As compared in Table 5 and Table 6, one partition is not sufficient to preserve the best quality. Too many partitions may reduce quality as well (as shown in the model with six partitions in Table 6). Empirically, a small number of equal partitions that naturally follow the structures of the models usually perform well. For example, ResNet-50 model is composed of four blocks, each sub-samples from the previous block. Thus a four-partition GaP is a natural choice. Transformer model contains an encoder and a decoder, with the decoder twice the number of attention blocks as the encoder. It is naturally divided to three partitions.

Table 5: The effect of different number of partitions for ResNet-50 models on ImageNet.

| Method | Sparsity Ratio | Distribution | Num. Partitions $\kappa$ | Top-1 Acc (%) |
|--------|---------------|--------------|--------------------------|---------------|
| *C*-GaP | 90% | uniform | 1 | 75.9 |
| *C*-GaP | 90% | uniform | 4 | 76.3 |

Table 6: The effect of different number of partitions for Transformer models on WMT-14 En-De.

| Method | Sparsity Ratio | Distribution | Num. Partitions $\kappa$ | SacreBLEU |
|--------|---------------|--------------|--------------------------|-----------|
| *C*-GaP | 90% | uniform | 1 | 26.8 |
| *C*-GaP | 90% | uniform | 3 | 27.7 |
| *C*-GaP | 90% | uniform | 6 | 27.1 |

## A.2  PARTITION THE MODEL IN A RANDOM MANNER

In the main text, each partition of the model consists of contiguous layers and they are cyclically grown and pruned. We also explore a random partition strategy for each step in the *C*-GaP for sparse ResNet-50 and Transformer models. Unlike the cyclic partition, a random partition chooses a uniformly random subset of layers to grow in each step of the GaP process. On the results for ResNet-50 described in Table 7, the random partition is slightly worse than the cyclic partition but still better than all previous existing solutions. On the results for Transformer described in Table 8, however, the accuracy gap is much larger. At 90% sparsity, Transformer model can achieve 27.7 SacreBLEU score in uniform sparsity with 3 partitions using C-GaP. When a random 3-partition is applied to a 90% sparsity Transformer, the SacreBLEU score is reduced to 27.0 and 26.9 for uniform and non-uniform sparsity, respectively. We conjecture that different model structures may result to different accuracy gaps. For ResNet-50, the correlations between neighboring layers may be small so each layer may be optimized independently. However, for Transformer, the inter-layer weight correlation is stronger (e.g., the $W_{key}$, $W_{query}$, $W_{value}$ in the same attention block have very strong correlation since they process the same input sequence for the self-attention computation)

This analysis shows that the partition mechanism affects the final model accuracy. Our scheduled GaP methods minimize the chance that the neighboring layers belong to different partitions, and have achieved superior accuracy.

Table 7: Different partition strategies for ResNet-50 models on ImageNet.

| Method | Sparsity Ratio | Distribution | Num. Partitions | Partition strategy | Top-1 Acc (%) |
|--------|---------------|--------------|-----------------|--------------------|--------------| 
| *C*-GaP | 90% | non-uniform | 4 | Cyclic | 77.9 |
| *C*-GaP | 90% | non-uniform | 4 | Random | 77.8 |

Table 8: Different partition strategies for Transformer models on WMT-14 En-De.

| Method | Sparsity Ratio | Distribution | Num. Partitions | Partition strategy | SacreBLEU |
|--------|---------------|--------------|-----------------|--------------------|-----------|
| *C*-GaP | 90% | uniform | 3 | Cyclic | 27.7 |
| *C*-GaP | 90% | uniform | 3 | Random | 27.0 |
| *C*-GaP | 90% | non-uniform | 3 | Random | 26.9 |

## A.3 Generalization to other sparse granularity

We use unstructured sparsity throughout the paper. In this section, we demonstrate that the GaP method is also applicable to other coarse-grained sparse granularity such as block-wise sparsity. The unit for the block-wise sparsity in this experiment is $1 \times 8$, i.e., if the weight matrix is viewed as many $1 \times 8$ blocks, then the sparsity ratio of the matrix is defined as the ratio of *all-zero* blocks. It is often believed that the accuracy would be impacted severely when the block size is 8 or greater. Table 9 compares the SacreBleu scores of the sparse Transformer models obtained by applying the 3-partition cyclic *C*-GaP and the prune-from-dense methods. It is observed that the GaP method improves over the prune-from-dense method significantly (27.21 vs. 26.21). This further validates the benefits of mask updating strategy brought by the GaP method.

Table 9: Different sparsifying methods for the Transformer models with structured sparsity on WMT-14 En-De.

| Method | Sparsity Ratio | sparse granularity | Num. Partitions $\kappa$ | BLEU |
|--------|---------------|--------------------|--------------------------|------|
| prune-from-dense | 80% | block $1 \times 8$ | - | 26.1 |
| *C*-GaP | 80% | block $1 \times 8$ | 3 | 27.2 |

## A.4 Pruning the fully connected layer in the ResNet-50 model

In the main context, the last fully connected (FC) layer of the ResNet-50 model is kept dense in the uniform sparsity distribution since it contributes to less than 1% of the FLOPs. In Table 10, we perform additional experiments to supplement Table 1 by pruning the last FC layer using the *C*-GaP method and comparing them with Evci et al. (2020). The results in Table 10 and Table 1 indicate that the accuracy of the ResNet-50 models with sparse and dense FC layers are similar, and both of them outperform the state-of-the-art results in Evci et al. (2020). Please note that models with the non-uniform sparsifying distribution in Table 1 already have the last FC layer pruned, thus the experiment setup is the same as the ones in Evci et al. (2020).

Table 10: Results of sparse ResNet-50 models on ImageNet.

| Method | Distribution | Sparsity ratio: 80% | | Sparsity ratio: 90% | |
|--------|-------------|---------------------|---------------|---------------------|---------------|
| | | Top-1 Acc (%) | FLOPS ($\times$e9) | Top-1 Acc (%) | FLOPS ($\times$e9) |
| RigL Evci et al. (2020) | uniform | 74.6 | 1.7 | 72.0 | 0.8 |
| RigL$_{5\times}$ Evci et al. (2020) | uniform | 76.6 | 1.7 | 75.7 | 0.8 |
| *C*-GaP (dense FC) | uniform | **77.9** | 1.7 | **76.3** | 0.8 |
| *C*-GaP (sparse FC) | uniform | **77.8** | 1.7 | **76.4** | 0.8 |

## B Experiment Setup

Most training scripts and hyper-parameters are inherited from NVIDIA (2020a) and Yan (2019). Identical hyper-parameters are applied to each GaP step and the final fine-tune stage. For example, the initial learning rate of each step is restored to the same value. This hyper-parameter restoration avoids pre-determining the total number of epochs before training and enables the GaP methods to continuously search for sparse masks.

### B.1 Image classification: ResNet-50 on ImageNet

In the image classification task, we use standard data augmentation, a batch size of 2048, cosine annealing learning rate schedule, SGD optimizer with a momentum of 0.875, and a weight decay of 3.05e-5. The learning rate is scheduled with a linear warm-up for 2 epochs before reaching the initial learning rate of 2.048. Each GaP step with non-uniform and uniform sparsity includes 30 of training, respectively. The final fine-tuning includes 150 epochs. For $C$-GaP, we train for 28 steps (i.e., 7 rounds with 4 partitions), and for $P$-GaP, we train for 32 steps (i.e., 8 rounds with 4 partitions). After the GaP step, we prune the dense partition(s) and fine-tune the model.

For the prune-from-dense method, we use iterative ADMM pruning. We first pretrain a dense model using 250 epochs, and perform ADMM regularization training for 250 epochs. Then we prune the model to 80% sparsity and finetune for another 250 epochs. Thus, the total number of epochs for 80% sparsity is 750 epochs. Starting from the 80% sparse model, we iteratively prune to 90% sparsity by firstly training the 80% sparse model with ADMM regularization for 250 epochs, and then prune to 90% sparsity and finetune 250 epochs. It takes a total of 1250 epochs to train a 90% sparsity model.

### B.2 Our Implementation Details of RigL

In this section, we provide the implementation details of the RigL (Evci et al., 2020). For the experiments with 100 training epochs, Most of the important hyper-paramters can be found in Evci et al. (2020). We adopt the settings in our PyTorch code base and the accuracy of RigL in 80% and 90% sparsity can be reproduced, both in uniform and ERK sparsity distributions, respectively. As shown in Table 1, our implementation has slightly higher accuracy, which indicates that our code base in PyTorch is comparable with the original implementation in Evci et al. (2020), and can be used for reproducing and extending the training of RigL without causing fairness issue.

Based on the above implementation, we reproduce the RigL results with 500 training epochs. However, Evci et al. (2020) doesn't provide detailed hyper-parameter settings for 500-epoch training. We perform extensive experiments and find out that using the same step learning rate decay scheduler can not reproduce the accuracy claimed in the original RigL paper, regardless of the learning rate decay factor or decay epochs. We change the step learning rate decay scheduler to the *cosine annealing* learning rate scheduler (Loshchilov & Hutter, 2017), and add *mixup* technique (Zhang et al., 2018a) with the hyper-parameter 0.2 to improve the generalization ability of the network. We find that the 500-epoch accuracy can be reproduced in these settings. We conjecture that the cosine annealing learning rate schedule and the mixup technique reconcile the overfitting problem that easily occurs in long period training. We inherit the RigL training recipe used in our 500-epoch training, and extend the training to 1200 epochs. The reproduced results are shown in Table 1.

### B.3 Object detection: SSD on COCO-2017

To train SSD on COCO-2017 dataset, we use an input size of $300 \times 300$ pixels, a batch size of 64, step learning rate schedule, SGD optimizer with a momentum of 0.9, and a weight decay of 5e-4. The learning rate is 2.6e-3 with 300 iterations of linear warm-up. Each GaP step includes 32 epochs of training and the final fine-tuning includes 65 epochs.

For the prune-from-dense method, we report the mAP of 24.3 by iteratively pruning. We first pretrain a dense model using 65 epochs, then we prune iteratively to $50\% \rightarrow 70\% \rightarrow 80\% \rightarrow 87.5\% \rightarrow 90\%$. For each pruning iteration, we train the sparse model for 65 epochs. It takes 390 epochs to get the final 90% sparsity model. Please note that directly pruning a 90% sparsity model only results to 21.98 mAP, which is significantly lower than the 24.3 mAP using the iterative pruning method. The model pruned using $C$-GaP method can achieve 25.9 mAP.

### B.4 3D object part segmentation: PoineNet++ on ShapeNet

When training PointNet++ (Qi et al., 2017; Yan, 2019) model on ShapeNet (Yi et al., 2016) dataset, we use a batch size of 32, step learning rate schedule with an initial learning rate of 0.001, SGD optimizer with a momentum of 0.9, and a weight decay of 1e-4. We pretrain a dense model using 250 epochs. Each step of GaP includes 25 epochs of training. The final fine-tuning includes 250 epochs.

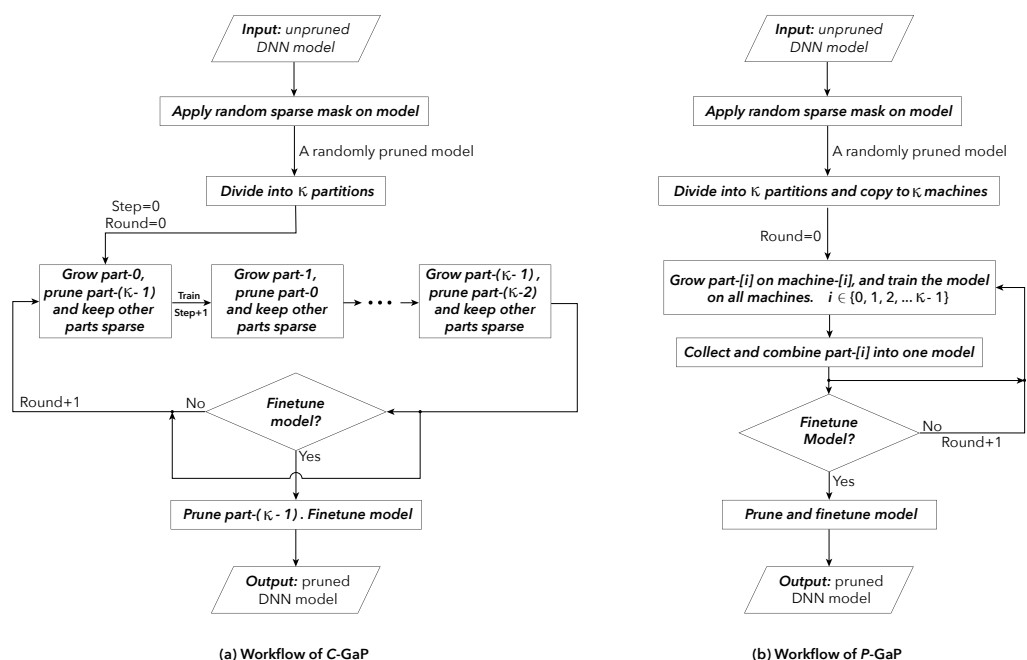

Figure 4: Training workflow of the (a) *C*-GaP and (b) *P*-GaP. In *C*-GaP (left), the model is partitioned and trained on one machine for multiple rounds. In each round, the sparse mask for each partition is grown and pruned cyclically. Each round has $\kappa$ steps when the model is partitioned into $\kappa$ parts. In *P*-GaP (right), a randomly initialized and sparsified model is copied and distributed to $\kappa$ machines when partitioned into $\kappa$ parts. Each round contains one step since all machines train the corresponding copy in parallel. An optimized model can be collected from the process and pruned to the desired sparsity scheme, then finetuned to restore accuracy.

## B.5    TRANSLATION: TRANSFORMER ON WMT-14 EN-DE DATASET

We use a batch size of 5120, inverse square-root scheduler with an initial learning rate of 0.000846, and Adam optimizer. Each step in GaP includes 10 epochs of training, and the final fine-tuning includes 40 epochs. We pretrain a dense model using 40 epochs. For prune from dense method, we use one-shot magnitude-based pruning to prune Transformer directly to 80% and 90% sparsity.

## C    THE DIFFERENCE BETWEEN SCHEDULED GAP AND RANDOM SPARSE MASK EXPLORATION

In our scheduled GaP algorithm, we intend to use all information in a model. i.e., explore all weights efficiently. One may argue that when trained sufficiently long, all weights will be explored even for random mask exploration. However, random sparse mask exploration and our scheduled GaP methods are like sampling operations with replacement and without replacement (Basu, 1958), and their required training time to get the same weight coverage differs a lot .

If we consider a very small model with only 10 weights, and each time we only train one weight and keep the remaining weights the same. When we apply the scheduled GaP algorithm (without replacement), each weight is guaranteed to be trained after 10 steps (one step indicates training one weight). This is because the scheduled GaP alternates the weight to train. The already trained weight is guaranteed not to be re-trained again until all the remaining weights are trained. On the other hand, if a random weight is selected to train in each step (with replacement), it may take 29 steps to be fairly confident that all weights are trained. That is almost 3 times more training time than the scheduled GaP. Please note, in our example, we only have 10 weights. A real neural network contains millions of weights and it is more difficult to guarantee a full exploration.

---

**Algorithm 2:** *P*-GaP training flow.

---

**Input:** An $L$-layer model with uninitialized weight $\Theta$; pruning ratio $r$.

**Output:** An $L$-layer sparse model satisfying the target sparsity requirement.

1  Divide all layers into $\kappa$ partitions, denoted by $S_i, i \in \{0, 1, \ldots, \kappa - 1\}$.
2  Initialize $f(x; \Theta \odot m)$ with random weights $\Theta$ and random masks $m$ that satisfy the sparsity requirement.
3  $step = 0$
4  **while** $step < K$ **do**
5      Send identical copies of $[\Theta, m]$ to all $\kappa$ nodes and denote them by $\Theta(i)$ and $m(i)$ on the $i$-th node.
6      **parfor** $i \in \{0, \ldots, \kappa - 1\}$ **do**
7          Grow $S_i$ by updating the mask $m(i)^{S_i} \leftarrow \texttt{ArgGrowTo}(\Theta(i)^{S_i})$.
8          Train the model $f(x; \Theta(i) \odot m(i))$ for $T$ epochs.
9      Collect the dense partition $\Theta(i)^{S_i}$ from the $i$-th machine.
10      Combine all $\Theta(i)^{S_i}$ into a dense model $\Theta^{S_0 \cup \cdots \cup S_{\kappa-1}}$ with the updated weights.
11      Prune $\Theta$ and update $m$ by $[\Theta, m] \leftarrow \texttt{ArgPruneTo}(\Theta, r)$.
12      $step = step + 1$.
13  Fine-tune $f(x; \Theta \odot m)$ for $T'$ epochs.

---

# D    A DETAILED WORKFLOW OF THE CYCLIC GAP AND PARALLEL GAP

In this section, we show the general flow of parallel GaP in Algorithm 2, and demonstrate the detailed workflow of the proposed cyclic GaP and parallel GaP method in Figure 4 as a supplement to the Algorithm 1 and Algorithm 2.

# E    PROOF OF PROPOSITION 1

## E.1   PROBLEM FORMULATION

Training deep neural networks can be formulated into minimizing the following problem:

$$\min_{\Theta \in \mathbb{R}^d} \quad F(\Theta) = \mathbb{E}_{x \sim D}[f(x; \Theta)] \tag{2}$$

where $\Theta \in \mathbb{R}^d$ is the model parameter to be learned, and $f(x; \Theta)$ is a smooth non-convex cost function in terms of $\Theta$. The random variable $x$ denotes the data samples that follow the distribution $D$. $\mathbb{E}_{x \sim D}[\cdot]$ denotes the expectation over the random variable $x$.

## E.2   NOTATION

We first clarify the notions used in the convergence analysis. To characterize how GaP updates model parameters in a partition-wise manner, we introduce matrices $U_i \in \mathbb{R}^{d \times d_i}, i = 1, \cdots, \kappa$, for which the identity matrix $I_d \in \mathbb{R}^{d \times d}$ can be represented as

$$I_d = (U_1, U_2, \cdots, U_\kappa). \tag{3}$$

- $\Theta^{S_i} = U_i^T \Theta \in \mathbb{R}^{d_i}$ is the model parameter at the $i$-th partition (i.e., partition $S_i$).

- $m^{S_i} \in \mathbb{R}^{d_i}$ is the mask for the $i$-th partition.

- $\nabla f(x; \Theta) \in \mathbb{R}^d$ and $\nabla F(\Theta) \in \mathbb{R}^d$ are the complete stochastic and accurate gradients in terms of $\Theta$, respectively.

- $\nabla_i f(x; \Theta) = U_i^T \nabla f(x; \Theta) \in \mathbb{R}^{d_i}$ is the stochastic gradient for the $i$-th partition

- $\nabla_i F(\Theta) = U_i^T \nabla F(\Theta) \in \mathbb{R}^{d_i}$ is the gradient for the $i$-th partition.

- $\Theta_{q,t}^{(i)} \in \mathbb{R}^d$ denotes the complete model parameter in which its $i$-th partition (i.e., $[\Theta_{q,t}^{(i)}]^{S_i} \in \mathbb{R}^{d_i}$) is newly updated at the $q$-th round and $t$-th inner iteration while the other partitions remain unchanged.

- $m_q^{(i)} \in \mathbb{R}^d$ is a complete mask vector for the $i$-th partition in the $q$-th round.

- $x_{q,t}^{(i)}$ is the data sampled at $q$-th round and $t$-th inner iteration to update the $i$-th partition.

### E.3 ALGORITHM REFORMULATION

In this subsection we reformulate the cyclic GaP (*C*-Gap) in Algorithm 1 in a way that can present the convergence analysis more easily. With the notations in Sec. E.2, the *C*-Gap Algorithm 1 can be reformulated as

---

**Algorithm 3:** *C*-GaP Algorithm (A math-friendly version)

---
1   Initialize $\Theta_{1,0}^{(1)}$ and $m_1^{(0)}$ randomly. Initialize pruning ratio $r$.

2   **for** *round* $q = 1, 2, ..., Q$ **do**

3      **for** *partition* $i = 1, 2, ..., \kappa$ **do**

4          Generate a mask $m_q^{(i)} = \text{GaPMask}(\Theta_{q,0}^{(i)}, m_q^{(i-1)}, r)$;

5          **for** $t = 1, ..., T$ **do**

6              Update $\Theta_{q,t}^{(i)} = \Theta_{q,t-1}^{(i)} - \gamma U_i \nabla_i f(x_{q,t-1}^{(i)}; \Theta_{q,t-1}^{(i)} \odot m_q^{(i)})$

7          Update $\Theta_{q,0}^{(i+1)} = \Theta_{q,T}^{(i)}$

8      Update $\Theta_{q+1,0}^{(1)} = \Theta_{q,T}^{(\kappa+1)}$ and $m_{q+1}^{(0)} = m_q^{(\kappa)}$;

9   **Output:** $\Theta_{Q,T}^{(\kappa)} \odot m_Q^{(\kappa)}$.

---

In $\text{GaPMask}(\Theta_{q,0}^{(i)}, m_q^{(i-1)}, r)$, the partition of $[m_q^{(i)}]^{S_i}$ is updated as:

> Prune (*i*-1)-th partition:   update $[m_q^{(i)}]^{S_{i-1}} \leftarrow \text{ArgPruneTo}([\Theta_{q,t}^{(i)}]^{S_{i-1}}, r)$ for pruning ratio $r$
>
>     Grow *i*-th partition:   Let $[m_q^{(i)}]^{S_i} = \mathbb{1}_{d_i} \in \mathbb{R}^{d_i}$
>
>     Other *j*-th partition:   $[m_q^{(i)}]^{S_j} = [m_q^{(i-1)}]^{S_j}$    (for all $j \neq i$ and $j \neq i - 1$)

and $m_q^{(i)} = [m_q^{(i)}]^{S_1 \cup S_2 \cup \cdots \cup S_\kappa}$.

### E.4 ASSUMPTIONS

We now introduce several assumptions on the cost function and the gradient noise, which are standard in the literature.

**Assumption 1** (SMOOTHNESS). *We assume the cost function $F(\Theta)$ is partition-wise L-smooth, i.e.,*

$$\|\nabla_i F(\Theta + U_i h_i) - \nabla_i F(\Theta)\| \leq L\|h_i\|, \quad \forall h_i \in \mathbb{R}^{d_i}. \tag{4}$$

*The above assumption implies that*

$$\|\nabla F(\Theta) - \nabla F(\Phi)\| \leq L\|\Theta - \Phi\|, \quad \forall \Theta, \Phi \in \mathbb{R}^d, \tag{5}$$

*or equivalently,*

$$F(\Theta) - F(\Phi) \leq \langle \nabla F(\Phi), \Theta - \Phi \rangle + \frac{L}{2}\|\Theta - \Phi\|^2, \quad \forall \Theta, \Phi \in \mathbb{R}^d, \tag{6}$$

**Assumption 2** (GRADIENT NOISE). *We assume for any k, t, and i that*

$$\mathbb{E}[\nabla_i f(x_{q,t}^{(i)}; \Theta)] = \nabla_i F(\Theta), \tag{7}$$

$$\mathbb{E}[\|\nabla_i f(x_{q,t}^{(i)}; \Theta) - \nabla_i F(\Theta)\|^2] \leq \sigma^2, \tag{8}$$

*where $\sigma > 0$ is a constent. Moreover, we assume the data sample $x_{q,t}^{(i)}$ is independent of each other for any k, t, and i.*

This assumption implies that the stochastic partition-gradient is unbiased and has bounded variance.

**Assumption 3** (MASK-INCURRED ERROR). *It is assumed that*

$$\|\Theta_{q,t-1}^{(i)} \odot m_q^{(i)} - \Theta_{q,t-1}^{(i)}\|^2 \leq \delta^2 \|\Theta_{q,t-1}^{(i)}\|^2 \quad where \quad \delta \in (0, 1). \tag{9}$$

Note that $[m_q^{(i)}]^{S_i} = \mathbb{1}_{d_i}$, and only the $i$-th partition in $\Theta_{q,t}^{(i)}$ is got updated during iterations $(q,1), \cdots, (q,T)$, we have

$$\|\Theta_{q,t-1}^{(i)} \odot m_q^{(i)} - \Theta_{q,t-1}^{(i)}\|^2 = \|\Theta_{q,0}^{(i)} \odot m_q^{(i)} - \Theta_{q,0}^{(i)}\|^2 \leq \delta^2 \|\Theta_{q,0}^{(i)}\|^2. \tag{10}$$

### E.5 CONVERGENCE ANALYSIS

Now we are ready to establish the convergence property of *C*-GaP in Algorithm 3. The arguments to establish the convergence are strait-forward:

- First, we need to establish the function value $\mathbb{E}[F(\Theta_{q,t-1}^{(i)})]$ will *decrease* to $\mathbb{E}[F(\Theta_{q,t}^{(i)})]$, up to an error term caused by the gradient noise and mask pruning, after each inner-iteration $t$ when updating the $i$-th partition (See Lemma 1).

- Based on the above fact, we next prove $\mathbb{E}[F(\Theta_{q,0}^{(i)})]$ will *decrease* to $\mathbb{E}[F(\Theta_{q,0}^{(i+1)})]$ when $T$-steps training of the $i$-th partition is completed (See Lemma 2).

- Finally, with the second fact, we show that $\mathbb{E}[F(\Theta_{q,0}^{(1)})]$ will *decrease* to $\mathbb{E}[F(\Theta_{q+1,0}^{(1)})]$ after the $k$-th outer round (See Lemmas 3 and 4). Since the function value decreases for each round $k$, we can prove that *C*-GaP will converge to a stationary solution after sufficiently large $Q$ rounds (See Theorem 1).

Next we present the detail analysis.

**Lemma 1** (DESCENT LEMMA AFTER EACH INNER-ITERATION). *Under Assumptions 1-3, it holds for each $k$, $i$, and $t$ that*

$$\mathbb{E}[F(\Theta_{q,t}^{(i)})] \leq \mathbb{E}[F(\Theta_{q,t-1}^{(i)})] - \frac{\gamma}{3}\mathbb{E}\|\nabla_i F(\Theta_{q,t-1}^{(i)})\|^2 + \frac{\gamma^2 L \sigma^2}{2} + \frac{2\gamma L^2 \delta^2}{3}\mathbb{E}\|\Theta_{q,0}^{(i)}\|^2. \tag{11}$$

**Remark 4.** *It is observed in Lemma 1 that the function value will decrease by $\frac{\gamma}{3}\mathbb{E}\|\nabla_i F(\Theta_{q,t-1}^{(i)})\|^2$ for each inner-iteration $t$. But such decrement suffers from two error terms: one is caused by stochastic gradient noise, and the other is by inexact pruning.*

*Proof.* Since $F(\Theta)$ is $L$-smooth (see Assumption 1), it holds that

$$F(\Theta_{q,t}^{(i)}) \leq F(\Theta_{q,t-1}^{(i)}) + \langle \nabla F(\Theta_{q,t-1}^{(i)}), \Theta_{q,t}^{(i)} - \Theta_{q,t-1}^{(i)} \rangle + \frac{L}{2}\|\Theta_{q,t}^{(i)} - \Theta_{q,t-1}^{(i)}\|^2$$

$$= F(\Theta_{q,t-1}^{(i)}) - \gamma \langle \nabla_i F(\Theta_{q,t-1}^{(i)}), \nabla_i f(x_{q,t-1}^{(i)}; \Theta_{q,t-1}^{(i)} \odot m_q^{(i)}) \rangle$$

$$+ \frac{\gamma^2 L}{2}\|\nabla_i f(x_{q,t-1}^{(i)}; \Theta_{q,t-1}^{(i)} \odot m_q^{(i)})\|^2 \tag{12}$$

With Assumption 2, it is easy to verify that

$$\mathbb{E}\langle \nabla_i F(\Theta_{q,t-1}^{(i)}), \nabla_i f(x_{q,t-1}^{(i)}; \Theta_{q,t-1}^{(i)} \odot m_q^{(i)}) \rangle$$

$$= \mathbb{E}_{\Theta_{q,t-1}^{(i)}} \left\{ \mathbb{E}[\langle \nabla_i F(\Theta_{q,t-1}^{(i)}), \nabla_i f(x_{q,t-1}^{(i)}; \Theta_{q,t-1}^{(i)} \odot m_q^{(i)}) \rangle | \Theta_{q,t-1}^{(i)}] \right\}$$

$$\overset{equation\ 7}{=} \mathbb{E}\langle \nabla_i F(\Theta_{q,t-1}^{(i)}), \nabla_i F(\Theta_{q,t-1}^{(i)} \odot m_q^{(i)}) \rangle. \tag{13}$$

Using a similar way, we can prove, with equation 8, that

$$\mathbb{E}\|\nabla_i f(x_{q,t-1}^{(i)}; \Theta_{q,t-1}^{(i)} \odot m_q^{(i)})\|^2 \leq \mathbb{E}\|\nabla_i F(\Theta_{q,t-1}^{(i)} \odot m_q^{(i)})\|^2 + \sigma^2 \tag{14}$$

By taking the expectation over equation 12 and substituting equation 13 and equation 14 into equation 12, we achieve

$$\mathbb{E}[F(\Theta_{q,t}^{(i)})] \leq \mathbb{E}[F(\Theta_{q,t-1}^{(i)})] - \gamma \mathbb{E}\langle \nabla_i F(\Theta_{q,t-1}^{(i)}), \nabla_i F(\Theta_{q,t-1}^{(i)} \odot m_q^{(i)}) \rangle$$

$$+ \frac{\gamma^2 L}{2}\mathbb{E}\|\nabla_i F(\Theta_{q,t-1}^{(i)} \odot m_q^{(i)})\|^2 + \frac{\gamma^2 L \sigma^2}{2}$$

$$= \mathbb{E}[F(\Theta_{q,t-1}^{(i)})] - \gamma \mathbb{E}\|\nabla_i F(\Theta_{q,t-1}^{(i)})\|^2 + \frac{\gamma^2 L}{2}\mathbb{E}\|\nabla_i F(\Theta_{q,t-1}^{(i)} \odot m_q^{(i)})\|^2 + \frac{\gamma^2 L \sigma^2}{2}$$

$$- \gamma \mathbb{E} \langle \nabla_i F(\Theta_{q,t-1}^{(i)}), \nabla_i F(\Theta_{q,t-1}^{(i)} \odot m_q^{(i)}) - \nabla_i F(\Theta_{q,t-1}^{(i)}) \rangle. \tag{15}$$

Note that

$$\mathbb{E}\|\nabla_i F(\Theta_{q,t-1}^{(i)} \odot m_q^{(i)})\|^2$$
$$\leq 2\mathbb{E}\|\nabla_i F(\Theta_{q,t-1}^{(i)})\|^2 + 2\mathbb{E}\|\nabla_i F(\Theta_{q,t-1}^{(i)}) - \nabla_i F(\Theta_{q,t-1}^{(i)} \odot m_q^{(i)})\|^2$$
$$\leq 2\mathbb{E}\|\nabla_i F(\Theta_{q,t-1}^{(i)})\|^2 + 2\mathbb{E}\|\nabla F(\Theta_{q,t-1}^{(i)}) - \nabla F(\Theta_{q,t-1}^{(i)} \odot m_q^{(i)})\|^2$$
$$\overset{equation\ 5}{\leq} 2\mathbb{E}\|\nabla_i F(\Theta_{q,t-1}^{(i)})\|^2 + 2L^2\mathbb{E}\|\Theta_{q,t-1}^{(i)} \odot m_q^{(i)} - \Theta_{q,t-1}^{(i)}\|^2$$
$$\overset{equation\ 10}{\leq} 2\mathbb{E}\|\nabla_i F(\Theta_{q,t-1}^{(i)})\|^2 + 2L^2\delta^2\mathbb{E}\|\Theta_{q,0}^{(i)}\|^2 \tag{16}$$

and, similarly,

$$- \mathbb{E}\langle \nabla_i F(\Theta_{q,t-1}^{(i)}), \nabla_i F(\Theta_{q,t-1}^{(i)} \odot m_q^{(i)}) - \nabla_i F(\Theta_{q,t-1}^{(i)}) \rangle$$
$$\leq \frac{1}{2}\mathbb{E}\|\nabla_i F(\Theta_{q,t-1}^{(i)})\|^2 + \frac{1}{2}\mathbb{E}\|\nabla_i F(\Theta_{q,t-1}^{(i)} \odot m_q^{(i)}) - \nabla_i F(\Theta_{q,t-1}^{(i)})\|^2$$
$$\leq \frac{1}{2}\mathbb{E}\|\nabla_i F(\Theta_{q,t-1}^{(i)})\|^2 + \frac{L^2\delta^2}{2}\mathbb{E}\|\Theta_{q,0}^{(i)}\|^2 \tag{17}$$

Substituting equation 16 and equation 17 into equation 15, and by setting $\gamma \leq 1/(6L)$, we have

$$\mathbb{E}[F(\Theta_{q,t}^{(i)})] \leq \mathbb{E}[F(\Theta_{q,t-1}^{(i)})] - \frac{\gamma(1-2\gamma L)}{2}\mathbb{E}\|\nabla_i F(\Theta_{q,t-1}^{(i)})\|^2 + \frac{\gamma^2 L\sigma^2}{2}$$
$$+ \frac{\gamma L^2\delta^2(1+2\gamma L)}{2}\mathbb{E}\|\Theta_{q,0}^{(i)}\|^2$$
$$\leq \mathbb{E}[F(\Theta_{q,t-1}^{(i)})] - \frac{\gamma}{3}\mathbb{E}\|\nabla_i F(\Theta_{q,t-1}^{(i)})\|^2 + \frac{\gamma^2 L\sigma^2}{2} + \frac{2\gamma L^2\delta^2}{3}\mathbb{E}\|\Theta_{q,0}^{(i)}\|^2. \tag{18}$$
$$\square$$

**Lemma 2** (DESCENT LEMMA AFTER EACH PARTITION UPDATE). *Under Assumptions 1-3, it holds for each $k$ and $i$ that*

$$\mathbb{E}[F(\Theta_{q,0}^{(i)})] \leq \mathbb{E}[F(\Theta_{q,0}^{(i+1)})] - \frac{\gamma}{3}\sum_{t=1}^{T}\mathbb{E}\|\nabla_i F(\Theta_{q,t-1}^{(i)})\|^2 + \frac{\gamma^2 L\sigma^2 T}{2} + \frac{2\gamma L^2\delta^2 T}{3}\mathbb{E}\|\Theta_{q,0}^{(i)}\|^2 \tag{19}$$

*Proof.* Summing the inequality over equation 11 for $t = 1, \cdots, T$, we achieve

$$\frac{\gamma}{3}\sum_{t=1}^{T}\mathbb{E}\|\nabla_i F(\Theta_{q,t-1}^{(i)})\|^2 \leq \sum_{t=1}^{T}\mathbb{E}[F(\Theta_{q,t-1}^{(i)}) - F(\Theta_{q,t}^{(i)})] + \frac{\gamma^2 L\sigma^2 T}{2} + \frac{2\gamma L^2\delta^2 T}{3}\mathbb{E}\|\Theta_{q,0}^{(i)}\|^2$$
$$= \mathbb{E}[F(\Theta_{q,0}^{(i)}) - F(\Theta_{q,T}^{(i)})] + \frac{\gamma^2 L\sigma^2 T}{2} + \frac{2\gamma L^2\delta^2 T}{3}\mathbb{E}\|\Theta_{q,0}^{(i)}\|^2 \tag{20}$$

which completes the proof. $\square$

**Lemma 3** (DESCENT LEMMA AFTER EACH ROUND). *Under Assumptions 1-3, it holds for each $k$ that*

$$\mathbb{E}[F(\Theta_{q,0}^{(1)})] \leq \mathbb{E}[F(\Theta_{q+1,0}^{(1)})] - \frac{\gamma}{3}\sum_{i=1}^{\kappa}\sum_{t=1}^{T}\mathbb{E}\|\nabla_i F(\Theta_{q,t-1}^{(i)})\|^2$$
$$+ \frac{\gamma^2 L\sigma^2 T\kappa}{2} + \frac{2\gamma L^2\delta^2 T}{3}\sum_{i=1}^{\kappa}\mathbb{E}\|\Theta_{q,0}^{(i)}\|^2 \tag{21}$$

*Proof.* Summing the inequality in equation 19 over $i = 1, \cdots, \kappa$, we have

$$\sum_{i=1}^{\kappa} \sum_{t=1}^{T} \mathbb{E}\|\nabla_i F(\Theta_{q,t-1}^{(i)})\|^2 \leq \frac{3}{\gamma} \sum_{i=1}^{\kappa} \mathbb{E}[F(\Theta_{q,0}^{(i)}) - F(\Theta_{q,T}^{(i)})] + \frac{3\gamma L\sigma^2 T\kappa}{2} + 2L^2\delta^2 T \sum_{i=1}^{\kappa} \mathbb{E}\|\Theta_{q,0}^{(i)}\|^2 \tag{22}$$

On the other hand, by the updating rule of $\Theta_{q,0}^{(i)}$, we have

$$\begin{aligned}
\sum_{i=1}^{\kappa} \mathbb{E}[F(\Theta_{q,0}^{(i)}) - F(\Theta_{q,T}^{(i)})] &= \mathbb{E}[F(\Theta_{q,0}^{(\kappa)}) - F(\Theta_{q,T}^{(\kappa)})] + \sum_{i=1}^{\kappa-1} \mathbb{E}[F(\Theta_{q,0}^{(i)}) - F(\Theta_{q,T}^{(i)})] \\
&= \mathbb{E}[F(\Theta_{q,0}^{(\kappa)}) - F(\Theta_{q+1,0}^{(1)})] + \sum_{i=1}^{\kappa-1} \mathbb{E}[F(\Theta_{q,0}^{(i)}) - F(\Theta_{q,0}^{(i+1)})] \\
&= \mathbb{E}[F(\Theta_{q,0}^{(1)}) - F(\Theta_{q+1,0}^{(1)})]. 
\end{aligned} \tag{23}$$

Substituting equation 23 into equation 22, we achieve

$$\begin{aligned}
&\sum_{i=1}^{\kappa} \sum_{t=1}^{T} \mathbb{E}\|\nabla_i F(\Theta_{q,t-1}^{(i)})\|^2 \\
&\leq \frac{3}{\gamma} \left( \mathbb{E}[F(\Theta_{q,0}^{(1)}) - F(\Theta_{q+1,0}^{(1)})] \right) + \frac{3\gamma L\sigma^2 T\kappa}{2} + 2L^2\delta^2 T \sum_{i=1}^{\kappa} \mathbb{E}\|\Theta_{q,0}^{(i)}\|^2
\end{aligned} \tag{24}$$

which completes the proof. $\qquad\square$

To finish the convergence proof, we still need to establish the relation between the decrement $\sum_{i=1}^{\kappa} \sum_{t=1}^{T} \mathbb{E}\|\nabla_i F(\Theta_{q,t-1}^{(i)})\|^2$ and $\mathbb{E}\|\nabla F(\Theta_{q,0}^{(1)})\|$ so that

**Lemma 4** (DESCENT LEMMA II AFTER EACH ROUND). *Under Assumptions 1-3, and learning rate $\gamma \leq \frac{1}{4\kappa LT}$, it holds for each $k$ that*

$$\mathbb{E}[F(\Theta_{q,0}^{(1)})] \leq \mathbb{E}[F(\Theta_{q+1,0}^{(1)})] - \frac{\gamma T}{12} \mathbb{E}\|\nabla F(\Theta_{q,0}^{(1)})\|^2 + \frac{\gamma^2 L\sigma^2 T^2\kappa}{6} + \frac{2\gamma L^2\delta^2 T^2}{3} \sum_{i=1}^{\kappa} \mathbb{E}\|\Theta_{q,0}^{(i)}\|^2 \tag{25}$$

*Proof.* In the following we lower bound the term $\sum_{i=1}^{\kappa} \sum_{t=1}^{T} \mathbb{E}\|\nabla_i F(\Theta_{q,t-1}^{(i)})\|^2$ in equation 21. Recall Algorithm 3 that

$$\Theta_{q,0}^{(i+1)} = \Theta_{q,0}^{(i)} - \gamma U_i \sum_{t=1}^{T} \nabla_i f(x_{q,t-1}^{(i)}; \Theta_{q,t-1}^{(i)} \odot m_q^{(i)}), \tag{26}$$

$$\Theta_{q,t}^{(i)} = \Theta_{q,0}^{(i)} - \gamma U_i \sum_{s=1}^{t} \nabla_i f(x_{q,s-1}^{(i)}; \Theta_{q,s-1}^{(i)} \odot m_q^{(i)}). \tag{27}$$

With these relations, it holds for $i \geq 1$ and $t \geq 1$ that

$$\begin{aligned}
&\mathbb{E}\|\Theta_{q,t}^{(i)} - \Theta_{q,0}^{(1)}\|^2 \\
&= \mathbb{E}\|\Theta_{q,t}^{(i)} - \Theta_{q,0}^{(i)} + \Theta_{q,0}^{(i)} - \Theta_{q,0}^{(i-1)} + \cdots + \Theta_{q,0}^{(2)} - \Theta_{q,0}^{(1)}\|^2 \\
&\overset{(a)}{=} \mathbb{E}\|\Theta_{q,t}^{(i)} - \Theta_{q,0}^{(i)}\|^2 + \mathbb{E}\|\Theta_{q,0}^{(i)} - \Theta_{q,0}^{(i-1)}\|^2 + \cdots + \mathbb{E}\|\Theta_{q,0}^{(2)} - \Theta_{q,0}^{(1)}\|^2 \\
&\overset{(b)}{=} \gamma^2 \mathbb{E}\| \sum_{s=1}^{t} \nabla_i f(x_{q,s-1}^{(i)}; \Theta_{q,s-1}^{(i)} \odot m_q^{(i)})\|^2 + \gamma^2 \sum_{j=1}^{i-1} \mathbb{E}\| \sum_{t=1}^{T} \nabla_j f(x_{q,t-1}^{(j)}; \Theta_{q,t-1}^{(j)} \odot m_q^{(j)})\|^2 \\
&\overset{(c)}{\leq} \gamma^2 \mathbb{E}\| \sum_{s=1}^{t} \nabla F(\Theta_{q,s-1}^{(i)} \odot m_q^{(i)})\|^2 + \gamma^2 \sum_{j=1}^{i-1} \mathbb{E}\| \sum_{t=1}^{T} \nabla F(\Theta_{q,t-1}^{(j)} \odot m_q^{(j)})\|^2 + i\gamma^2\sigma^2
\end{aligned}$$

$$\overset{(d)}{\leq} t\gamma^2 \sum_{s=1}^{t} \mathbb{E}\|\nabla F(\Theta_{q,s-1}^{(i)} \odot m_q^{(i)})\|^2 + \gamma^2 T \sum_{j=1}^{i-1} \sum_{t=1}^{T} \mathbb{E}\|\nabla F(\Theta_{q,t-1}^{(j)} \odot m_q^{(j)})\|^2 + i\gamma^2 \sigma^2$$

$$\leq \gamma^2 T \sum_{j=1}^{i} \sum_{t=1}^{T} \mathbb{E}\|\nabla F(\Theta_{q,t-1}^{(j)} \odot m_q^{(j)})\|^2 + \kappa\gamma^2 \sigma^2$$

$$\leq 2\gamma^2 T \sum_{j=1}^{i} \sum_{t=1}^{T} \mathbb{E}\|\nabla F(\Theta_{q,t-1}^{(j)})\|^2 + 2\gamma^2 L^2 T \delta^2 \sum_{j=1}^{i} \sum_{t=1}^{T} \mathbb{E}\|\Theta_{q,0}^{(j)}\|^2 + \kappa\gamma^2 \sigma^2$$

$$= 2\gamma^2 T \sum_{j=1}^{i} \sum_{t=1}^{T} \mathbb{E}\|\nabla F(\Theta_{q,t-1}^{(j)})\|^2 + 2\gamma^2 L^2 T^2 \delta^2 \sum_{j=1}^{i} \mathbb{E}\|\Theta_{q,0}^{(j)}\|^2 + \kappa\gamma^2 \sigma^2 \qquad (28)$$

where (a) holds because $\Theta_{q,0}^{(i)} - \Theta_{q,0}^{(i-1)}$ and $\Theta_{q,0}^{(j)} - \Theta_{q,0}^{(j-1)}$ are orthogonal to each other when $i \neq j$, (b) holds because of equation 26 and equation 27, (c) holds because of Assumption 2, and (d) holds because of the Jensen's inequality. With the above inequality, it holds for $t \in [1, T]$ that

$$\mathbb{E}\|\nabla_i F(\Theta_{q,0}^{(1)})\|^2 \leq 2\mathbb{E}\|\nabla_i F(\Theta_{q,t}^{(i)})\|^2 + 2\mathbb{E}\|\nabla_i F(\Theta_{q,t}^{(i)}) - \nabla_i F(\Theta_{q,0}^{(1)})\|^2$$

$$\leq 2\mathbb{E}\|\nabla_i F(\Theta_{q,t}^{(i)})\|^2 + 2\mathbb{E}\|\nabla F(\Theta_{q,t}^{(i)}) - \nabla F(\Theta_{q,0}^{(1)})\|^2$$

$$\overset{equation\ 5}{\leq} 2\mathbb{E}\|\nabla_i F(\Theta_{q,t}^{(i)})\|^2 + 2L^2 \mathbb{E}\|\Theta_{q,t}^{(i)} - \Theta_{q,0}^{(1)}\|^2$$

$$\overset{equation\ 28}{\leq} 2\mathbb{E}\|\nabla_i F(\Theta_{q,t}^{(i)})\|^2 + 4L^2 \gamma^2 T \sum_{j=1}^{i} \sum_{t=1}^{T} \mathbb{E}\|\nabla F(\Theta_{q,t-1}^{(j)})\|^2$$

$$+ 4\gamma^2 L^4 T^2 \delta^2 \sum_{j=1}^{i} \mathbb{E}\|\Theta_{q,0}^{(j)}\|^2 + 2\kappa\gamma^2 \sigma^2 L^2 \qquad (29)$$

Summing the above inequality over $t = 0, 1, \cdots, T$, we have

$$T\mathbb{E}\|\nabla_i F(\Theta_{q,0}^{(1)})\|^2 \leq 2 \sum_{t=1}^{T} \mathbb{E}\|\nabla_i F(\Theta_{q,t-1}^{(i)})\|^2 + 4T^2 L^2 \gamma^2 \sum_{j=1}^{i} \sum_{t=1}^{T} \mathbb{E}\|\nabla F(\Theta_{q,t-1}^{(j)})\|^2$$

$$+ 4\gamma^2 L^4 T^3 \delta^2 \sum_{j=1}^{i} \mathbb{E}\|\Theta_{q,0}^{(j)}\|^2 + 2\kappa\gamma^2 \sigma^2 L^2 T \qquad (30)$$

Summing the above inequality over $i = 1, \cdots, \kappa$, we have

$$T \sum_{i=1}^{\kappa} \mathbb{E}\|\nabla_i F(\Theta_{q,0}^{(1)})\|^2 \leq 2 \sum_{t=1}^{T} \sum_{i=1}^{\kappa} \mathbb{E}\|\nabla_i F(\Theta_{q,t-1}^{(i)})\|^2 + 4T^2 L^2 \gamma^2 \kappa \sum_{j=1}^{\kappa} \sum_{t=1}^{T} \mathbb{E}\|\nabla F(\Theta_{q,t-1}^{(j)})\|^2$$

$$+ 4\gamma^2 L^4 T^3 \delta^2 \kappa \sum_{j=1}^{\kappa} \mathbb{E}\|\Theta_{q,0}^{(j)}\|^2 + 2\kappa^2 \gamma^2 \sigma^2 L^2 T$$

$$= \left(4T^2 L^2 \gamma^2 \kappa + 2\right) \sum_{t=1}^{T} \sum_{i=1}^{\kappa} \mathbb{E}\|\nabla_i F(\Theta_{q,t-1}^{(i)})\|^2$$

$$+ 4\gamma^2 L^4 T^3 \delta^2 \kappa \sum_{j=1}^{\kappa} \mathbb{E}\|\Theta_{q,0}^{(j)}\|^2 + 2\kappa^2 \gamma^2 \sigma^2 L^2 T \qquad (31)$$

Combining equation 24 and equation 31, we have

$$T \sum_{i=1}^{\kappa} \mathbb{E}\|\nabla_i F(\Theta_{q,0}^{(1)})\|^2$$

$$\leq \left(4T^2 L^2 \gamma^2 \kappa + 2\right) \left(\frac{3}{\gamma}\left(\mathbb{E}[F(\Theta_{q,0}^{(1)}) - F(\Theta_{q+1,0}^{(1)})]\right) + \frac{3\gamma L\sigma^2 T\kappa}{2} + 2L^2 \delta^2 T \sum_{i=1}^{\kappa} \mathbb{E}\|\Theta_{q,0}^{(i)}\|^2\right)$$

$$+ 4\gamma^2 L^4 T^3 \delta^2 \kappa \sum_{i=1}^{\kappa} \mathbb{E}\|\Theta_{q,0}^{(i)}\|^2 + 2\kappa^2\gamma^2\sigma^2 L^2 T$$

$$\overset{(a)}{\leq} \frac{12}{\gamma}\left(\mathbb{E}[F(\Theta_{q,0}^{(1)}) - F(\Theta_{q+1,0}^{(1)})]\right) + (6\gamma LT\kappa + 2\kappa^2\gamma^2 L^2 T)\sigma^2$$

$$+ \left(4\gamma^2 L^4 T^3 \delta^2 \kappa + 8L^2\delta^2 T\right) \sum_{i=1}^{\kappa} \mathbb{E}\|\Theta_{q,0}^{(i)}\|^2$$

$$\overset{(b)}{\leq} \frac{12}{\gamma}\left(\mathbb{E}[F(\Theta_{q,0}^{(1)}) - F(\Theta_{q+1,0}^{(1)})]\right) + 8\gamma LT\kappa\sigma^2 + 10L^2\delta^2 T \sum_{i=1}^{\kappa} \mathbb{E}\|\Theta_{q,0}^{(i)}\|^2 \tag{32}$$

where (a) and (b) use the facts that

$$2T^2 L^2 \gamma^2 \kappa + 1 \leq 2 \quad \text{(it is enough to set } \gamma \leq \tfrac{1}{\sqrt{2\kappa}LT}), \tag{33}$$

$$\kappa^2\gamma^2 L^2 T \leq \gamma LT\kappa \quad \text{(it is enough to set } \gamma \leq \tfrac{1}{\kappa L}). \tag{34}$$

Then the inequality equation 32 becomes

$$\mathbb{E}\|\nabla F(\Theta_{q,0}^{(1)})\|^2 \leq \frac{12}{\gamma T}\left(\mathbb{E}[F(\Theta_{q,0}^{(1)}) - F(\Theta_{q+1,0}^{(1)})]\right) + 8\gamma L\kappa\sigma^2 + 10L^2\delta^2 \sum_{i=1}^{\kappa} \mathbb{E}\|\Theta_{q,0}^{(i)}\|^2. \tag{35}$$

$\square$

Finally we establish the convergence of the *C*-Gap algorithm as follows:

**Theorem 1** (C-GAP CONVERGENCE). *Under Assumptions 1-3, if learning rate is set as $\gamma = \frac{1}{4\kappa LT\sqrt{Q}}$, it holds that*

$$\frac{1}{Q}\sum_{q=1}^{Q}\mathbb{E}\|\nabla F(\Theta_{q,0}^{(1)} \odot m_q^{(1)})\|^2 \leq \frac{G}{\sqrt{Q}} + \frac{10L^2 T\delta^2}{Q}\sum_{q=1}^{Q}\sum_{i=1}^{\kappa}\mathbb{E}\|\Theta_{q,0}^{(i)}\|^2 \tag{36}$$

*where $G = 96\kappa L\mathbb{E}[F(\Theta_{1,0}^{(1)})] + \sigma^2$ is a constant.*

*Proof.* Since

$$\mathbb{E}\|\nabla F(\Theta_{q,0}^{(1)} \odot m_q^{(1)})\|^2 \leq 2\mathbb{E}\|\nabla F(\Theta_{q,0}^{(1)})\|^2 + 2L^2\delta^2\mathbb{E}\|\Theta_{q,0}^{(1)}\|^2, \tag{37}$$

we thus have

$$\mathbb{E}\|\nabla F(\Theta_{q,0}^{(1)} \odot m_q^{(1)})\|^2$$

$$\overset{equation\ 25}{\leq} \frac{24}{\gamma T}\left(\mathbb{E}[F(\Theta_{q,0}^{(1)}) - F(\Theta_{q+1,0}^{(1)})]\right) + 8L^2 T\delta^2 \sum_{i=1}^{\kappa}\mathbb{E}\|\Theta_{q,0}^{(i)}\|^2 + 4\gamma LT\kappa\sigma^2 + 2L^2\delta^2\mathbb{E}\|\Theta_{q,0}^{(1)}\|^2$$

$$\leq \frac{24}{\gamma T}\left(\mathbb{E}[F(\Theta_{q,0}^{(1)}) - F(\Theta_{q+1,0}^{(1)})]\right) + 10L^2 T\delta^2 \sum_{i=1}^{\kappa}\mathbb{E}\|\Theta_{q,0}^{(i)}\|^2 + 4\gamma LT\kappa\sigma^2 \tag{38}$$

Summing the above inequality over $q = 1, \cdots, Q$ and taking its average, we achieve

$$\frac{1}{Q}\sum_{q=1}^{Q}\|\nabla F(\Theta_{q,0}^{(1)} \odot m_q^{(1)})\|^2 \leq \frac{24}{\gamma TQ}\mathbb{E}[F(\Theta_{1,0}^{(1)})] + \frac{10L^2 T\delta^2}{Q}\sum_{q=1}^{Q}\sum_{i=1}^{\kappa}\mathbb{E}\|\Theta_{q,0}^{(i)}\|^2 + 4\gamma LT\kappa\sigma^2 \tag{39}$$

If we let $\gamma = \frac{1}{4\kappa LT\sqrt{Q}}$, the above inequalities becomes

$$\frac{1}{Q}\sum_{q=1}^{Q}\|\nabla F(\Theta_{q,0}^{(1)} \odot m_q^{(1)})\|^2 \leq \frac{96\kappa L}{\sqrt{Q}}\mathbb{E}[F(\Theta_{1,0}^{(1)})] + \frac{\sigma^2}{\sqrt{Q}} + \frac{10L^2 T\delta^2}{Q}\sum_{q=1}^{Q}\sum_{i=1}^{\kappa}\mathbb{E}\|\Theta_{q,0}^{(i)}\|^2 \tag{40}$$

which completes the proof.

$\square$

