# OpenReview forum: "Effective Model Sparsification by Scheduled Grow-and-Prune Methods"
_ICLR.cc/2022/Conference — ICLR 2022 Poster_

### Official Review · Reviewer_Saw3 · 2021-10-16

**Correctness:** 4
**Technical Novelty And Significance:** 3
**Empirical Novelty And Significance:** 3
**Recommendation:** 8
**Confidence:** 5

**Main Review:**

I will go sequentially starting with general comments followed by strengths and weaknesses.

The paper is hard to read and would benefit from a revision from a writing perspective. The ideas can be further simplified while explanation (the verbatim has changed a bit from the NeurIPS version but it can still be implied). The illustrations are very helpful, but the text and the language often made me feel lost (the only reason, I am comfortable right now is due to the conversation with reviewers that made me understand more). I would strongly recommend the authors revise the write-up for the next version (again).

Strengths:

1) Figure 1 and Figure 2 are very very clear and basically give the idea of what is being done in the paper completely.
2) The motivation for GaP is well explained.
3) The pseudo code and method section is clear.
4) The idea is simple yet seems to be effective. It is sort of a greedy way (even though scheduled) to sparsify the network, where the dense portion is like dense gradient update of multiple methods in the literature (eg., RigL) but also the classical iterative hard thresholding. The idea makes sense, fits well into the greedy growing NAS literature.
5) Parallel Gap is a nice engineering solution to make use of parallelization of GPUs and shows that the costs saved are worth it. I really like the new figure illustrating P-GaP, this helps more people understand its potential.
6) The experimental results are good and move past the usual image classification benchmarks to object detection, 3D object part segmentation, and transformers.
7) The ablations on the # partitions are interesting.
8) The related work did improve from the previous version, I appreciate the authors for taking time and reviewing more papers to put things in perspective.
9) I am extremely happy to see the training costs as part of the tables. This was very much lacking in the previous revisions and puts things into perspective. The cost of training is a trade-off in this case which is at least presented to the readers. I have some issues with the costs involved which I will mention in the weaknesses, but this is very promising.

I am not an expert on theory so I will defer to other reviewers on that aspect. However, I don't think theory is needed to justify this paper, if this is a theory for the sake of theory, I would suggest moving it to the appendix and use the space to make compelling cases and use cases for the algorithm itself.

Weaknesses:
1) The major concern, similar to last time, is the computational costs involved. Is the trade-off worth it? I can see this paper as just pushing the limits on the inference end, getting the best models that are sparse, but the gain of 2% with say 800 additional epochs is something I am not sure how to weigh.
2) Table 1 is not informative at all, and space could be used for something else. This is a very subjective table, I have no idea what better or good means here. I don't think it is worthy of the space.
3) The non-uniform sparsity scheme for GaP is missing (again). This was explained to be done using global sparsity (in the rebuttal), but it is not present in the paper.
4) While I agree FLOPs definition in this paper is correct, taking a line to explain why previous papers were reporting it the incorrect way would be great.

Overall, the paper makes sense and has improved to an extent from the previous version as I mentioned above. However, I would like to have a discussion with the authors about the questions I mentioned above. I was also promised a "limitations and societal impact" section trying to explain the trade-offs and issues which was not delivered. I am looking forward to it.

I think the paper is publishable but can be made stronger after the rebuttal.


**Summary Of The Paper:**

Note: I have previously reviewed this paper for NeurIPS 2021 and I will be reusing my review from then and add the additional comments along the way based on how the paper has been updated along with the promised updates from the previous revisions.

This paper proposes a way to obtain sparse neural networks by repeatedly growing and pruning parts of the model in a scheduled fashion. Figure 1 gives a good idea of the whole technique. It basically divides the whole network into K parts and only one of these K parts is dense while the rest are sparse. The dense part keeps cycling ensuring over-parameterized learning when needed. The paper proposes that these scheduled routines help in improving the accuracy of the sparse neural network models compared to other sparsity-inducing techniques.

The paper has extensive experimental evaluation on tasks like classification, object detection, 3D object part segmentation, and Transformers. The authors also try to provide some theoretical explanations to these proposed ideas.

The paper has updated visualizations to make the algorithm much more clearer from the earlier version which is commendable.

**Summary Of The Review:**

I have reviewed this paper for NeurIPS 2021 and gave it a score of 6. But the paper had a bunch of issues making it not ready for publication.

This time, the authors tried to put things in perspective. The empirical performance of the method is strong but I would like to see the limitations of the method along with a discussion of trade-offs.

I am giving it a score of 6 but will be willing to increase it to 8 if the authors respond back with valid explanations for weaknesses.

---------------------------------------------
After rebuttal:

I am convinced about the contributions of the paper and I would be arguing for the acceptance of the same during the internal discussions. Increasing score to an 8.

---

> ### Author Response · Authors · 2021-11-15
> **Response to Reviewer Saw3**
>
> ## Response - Part 1/2
>
> Dear reviewer,
>
> We would like to sincerely thank you again for your valuable time and comments. We apologize for our prior oversights. We have addressed your questions in the revised paper PDF (changes are highlighted in blue).  to improve our work. Below are our responses.
>
> **Q1: About the writing.**
>
> A1: We greatly appreciate your suggestions.  Though we have revised quite a bit since we submitted the paper to NeurIPS, it is obvious to us now that our modifications still have not met your standard. We have done our best to modify the text to make the ideas clearer. However, with so many lines of text, we have a challenging time finding the confusing part.  It would be very helpful for us if you could suggest some detailed places that we could improve. Could you please let us know the texts in line numbers that confuse you? We have revised them in our paper PDF. Thanks a lot!
>
>
> **Q2: Long training time for high accuracy.**
>
> A2: Our proposed GaP method is to obtain the best quality of the pruned model under a runtime budget. In our perspective, spending a bit more time in training to find a better model is well rewarded since models may be deployed to thousands of devices and inference billions of times. Based on the researches in Facebook [r4-1, r4-2], the models are inferenced many billions of times per day. Thus, a better pruning algorithm resulting to a smaller model may save a lot more computation on the inference side. As an example, the 90% non-uniform sparsity ResNet-50 model pruned using GaP achieves 77.9 top-1 accuracy on ImageNet, outperforming the previous SOTA of RigL-12x at 80% non-uniform sparsity by 0.5%. It also saves over 50% inference FLOPs with less training cost. Even if it is compared with the model that is trained for ~100 epochs at 80% sparsity (with 1.4% less accuracy), the extra training cost (900 epochs X 1M images per epoch) is only a tiny fraction of the inference cost in one day. The 50% inference FLOPs savings is a pure gain if the model is deployed for multiple days.
>
> Put the computation saving aside, many top-tier internet companies (e.g. Google, Facebook) strive for ~0.01% accuracy gains on deployable recommendation models. With trillions of inferences every day, such accuracy gains translate to real revenues. Thus, a more efficient pruning algorithm could enable them to create better deployable models.
>
> Thus, we believe that our method advances the state-of-the-art in model compression efficiency.
>
>
>
> **Q3: Delete table 1**
>
> A3: Thank you very much! We have deleted Table 1 (in the old version) as the you suggested.
>
>
> **Q4: About how to obtain non-uniform sparsity.**
>
> A4: We apologize again for not integrating how we obtain the non-uniform sparsity in the paper. We intended to add this discussion in our paper but we missed it somehow. For non-uniform sparsity, we choose to prune weights based on their magnitude across all layers. That is, we set a global threshold for the entire model and prune any weight magnitude smaller than that. After pruning, one partition is grown to dense and the training continues. Finally, all layers are pruned non-uniformly by their weight magnitudes again and finetuned. We have added this part to our newly revised paper PDF and shown our revision to you.
>
>
> **Q5: Correct FLOPs.**
>
> A5: We would like to thank the reviewer for pointing this out. We have added a footnote with references to explain the FLOPs definition.
>
> **(Please continue to read part 2)**

---

> > ### Comment · Reviewer_Saw3 · 2021-11-17
> > **Thanks for the clarifications**
> >
> > I appreciate the authors for their detailed rebuttal. I am happy with the responses with Q3, Q4 (the new write-up in the paper helps), and Q5.
> >
> > Coming to Q1 (about writing). I know this is not optimal, but it is not certain parts of the paper but the writing as a whole that is slightly hard to parse. With the empirical strength of your paper and the novelty of the idea, I would expect you to make things more clear if possible to cater to as many people as you can. The new additions in the manuscript help a bit, but I recommend the authors give a fresh look in the next revision. I think I was lost in Sections 2 and 3 with the dense packing of information.
> >
> > Q2. I agree with your response and I am impressed with the updated results from NeurIPS submission for fair comparison as well. I would ask you to have the motivation of real-world deployment in the main paper and not in the appendix. The only aspect I don't agree with is the claim of 0.01% translating to real revenue, I would refrain from that claim because it is much more nuanced than that.
> >
> > Thanks for the limitations and societal impact section. It is a good start but I would also like for the authors to acknowledge the bias of sparse models. I would also love for this to be in the main paper and not in the appendix as I said earlier about the motivation as well.
> >
> > I am convinced about the contributions of the paper and I would be arguing for the acceptance of the same during the internal discussions.

---

> > > ### Author Response · Authors · 2021-11-19
> > > **Thanks for additional comments and suggestions**
> > >
> > > Dear reviewer,
> > >
> > > Thank you very much for your response and for raising the score. We are happy that our responses have addressed portions of your concerns.
> > >
> > > As for Q2, we have enriched the wording in the introduction section to motivate the real-world deployment scenarios. We are sorry for the claim of 0.01% accuracy gain translating to revenues. It is based on one model and may not be generalized to other models. It is not mentioned in the paper however.
> > >
> > > Thank you very much for raising the writing quality issue again. With your guidance, we do realize this time that our writing on the GaP methodology is indeed difficult for first time readers. Thus, we have comprehensively rewritten the methodology section and the related parts in the introduction section. We have also refined the wording in the experiments.
> > >
> > > We have also moved the limitation and societal impact section to the main paper, and have added the content discussing the bias of sparse models.
> > >
> > > We have uploaded the revised paper, and the revised portion is highlighted in purple. Please check it for details.
> > >
> > >
> > > Again, thank you very much for your valuable time and constructive suggestions to make our paper better. Please feel free to comment back if you have additional questions. We'd love to answer them.

---

> ### Author Response · Authors · 2021-11-15
> **Response to Reviewer Saw3**
>
> ## Response - Part 2/2
>
> **(Continue from part 1)**
>
> **Q6: The limitation and societal impact of this paper.**
>
> A6: Again, we sincerely apologize for not adding a section discussing limitations and societal impact of this paper. We have revised our paper and add this discussion in Appendix E. Below is the text for these sections.
>
> --------
> **Limitations:**
>
> In this work, we focus on the unstructured sparsity to demonstrate the algorithm-level innovations of our proposed GaP methods. We recognize that the structured sparsity is a natural extension of our work. However, we only preliminarily include a block sparsity example in Table 9 to demonstrate its applicability to structured sparsity and its potential to improve the inference speed. More detailed analysis is still a work in progress.
>
> The scheduled GaP methods require more training time in terms of the number of epochs than using some conventional pruning methodology, which may potentially limit its applicability in both academia and industry when the training compute resources are limited. Thus, the scheduled GaP methods are mostly beneficial to models sensitive to accuracy and runtime performance, and the benefit of the reduced inference time and/or improved accuracy out weigh the cost of the moderately longer training time.
>
> The key hyper-parameter settings such as partition numbers or partition strategies are determined heuristically. In this work, we try to divide the models to partitions with similar parameter counts or computation. We also intuitively group consecutive layers to the same partition, based on our hypothesis that adjacent layers are tighter correlated than more distant layers. With limited time and compute resources, our evaluation could not thoroughly cover all the hyper-parameter settings, so we explored the most relevant ones based on our experience.
>
> Our analysis in Proposition 1 cannot be directly extended to cover P-GaP since the analysis is built on the partition-wise cyclic updating structure of C-GaP. We will leave the analysis for P-GaP as a future work.
>
>
> **Societal Impact**
>
> The scheduled GaP methods target to obtain the best quality of the pruned model under a runtime budget. Based on the researches in Facebook [r4-1, r4-2], many models are inferenced many billions of times per day, which makes the cost of pruning such models a tiny fraction of the inference cost in one day. Thus, a better pruning algorithm resulting to a smaller model may save a lot more computation on the inference side, which is of high value for the community and society to achieve Green AI [r4-3]. On the other hand, our method cannot prevent the possible malicious usage, which may cause negative societal impact.
>
> ----------
>
>
>
> We greatly thank your valuable suggestions (twice), and we will revise our paper and make sure we address every point in your review. Please feel free to let us know if you have additional comments and we are always happy to discuss them here.
>
>
> [r4-1] Applied Machine Learning at Facebook: A Datacenter Infrastructure Perspective
>
> [r4-2] Machine Learning at Facebook: Understanding Inference at the Edge
>
> [r4-3] Schwartz, Roy, et al. "Green ai." Communications of the ACM, 2020

---

### Official Review · Reviewer_Zewc · 2021-11-01

**Correctness:** 4
**Technical Novelty And Significance:** 3
**Empirical Novelty And Significance:** 3
**Recommendation:** 6
**Confidence:** 4

**Main Review:**

Strengths:
1. This study provides one model compression method. With theoretical analysis, authors proved the correctness of this method.
2. This method shows preferable performance in constrast to prevent methods in various experiments.

Weakness:
1. The creativety of this method is somewhat weak. The pruning-rewiring-pruning method have already been used in several papers, such as "DSD: Dense-Sparse-Dense Training for Deep Neural Networks" and some papers in the references.
2. Batch normalization layer is used to compute the statistics in current network. Wheter this kind of layer should be managed specially?
3. In experiments, the epochs of this method is much larger, even $10$ times more than other methods, which has negative societal impact and will limit its  applications.
4. The sparsity is very high, under 80\% -90 \% sparsity, the model maintains high accuracy.  However, the FLOPs has minor relations with inference speed. Unstructured pruning methods have limited befits to the memory even considering the storage mode of sparse matrix. So, what's the real gain by this method?

**Summary Of The Paper:**

This study takes use of pruning-rewiring-pruning method to get a high unstructured sparse model. Broadly, this is one dropout regularization method. With step-by-step growing and pruning, most of the weights are close to $0$ and will be pruned in the next step. Experiments show that this training method can improve the sparsity and performance for a wide range of architectures and applications,

**Summary Of The Review:**

This study provides one new training method of sparse model by scheduled grow-and-prune (GaP) methodology. The authors provides  a series of experiments. However, this study requires additional comparions, particularly some moden pruning methods under same experiment settings, such as SCOP, HRank, etc. Alternatively, the authors should include more information about the gains of this method.

---

> ### Author Response · Authors · 2021-11-15
> **Response to Reviewer Zewc:**
>
> ## Response - Part 1/2
>
> Dear reviewer,
>
> Thank you for your time reviewing our paper. The questions and suggestions are all very constructive. We provide our responses below, and we have revised our paper PDF (changes are highlighted in blue).  Please feel free to let us know if you have additional comments and we are always happy to discuss them here.
>
> **Q1: Compare to DSD and similar papers.**
>
> A1: The GaP method is practically effective on model pruning and theoretically sound for the training convergence. It is different from the prior works, especially the sparse mask exploration methods (DeepR, SET, RigL, DSR and DPF in our paper) in the following aspects. 1) We introduce a scheduled partition method based on layers. We coordinately explore all weights in a layer. And we alternately explore the weights in different layers. 2) We guarantee all weights are explored equally in an efficient manner since we sample without replacement. Prior mask exploration methods sample with replacement and cannot guarantee exploring all weights in the training (Please refer to our answer to Q2 of xnEk for more detailed discussion). Experiments have also confirmed the superior quality of our method. Thus, we believe our method is novel.
>
> The goal of DSD is different from our paper. The DSD method is proposed to improve the accuracy of the dense models by applying a sparsity regularization and finetuning the re-initialized dense network. We target to improve the accuracy of the sparse models using a novel scheduled grow-and-prune method. Additionally, we have evaluated the DSD method for training a sparse model in Appendix A.1. For ResNet-50 on ImageNet (Table 5), we train the model using DSD for 2750 epochs (we alternate training the dense and sparse models for 250 epochs each, and we train a total of 11 steps), and the resulting sparse model only achieves 75.9% top-1 accuracy at 90% uniform sparsity. While in our paper, the C-GaP achieves 76.3% top-1 accuracy with a total of 990 training epochs. Similarly, the Transformer model trained using DSD (one partition) result to 0.9 SacreBLEU score loss compared with a 3-partition model trained using C-GaP (Table 6). We can observe from the results that partitions are important, and the GaP method obtains better sparse models than using the DSD method.
>
>
> **Q2: Batch normalization layer is used to compute the statistics in current network. Whether this kind of layer should be managed specially?**
>
> A2: We are confused about this question. We do not use BN to compute the statistics of the model. The only place we mention BN is in section 2.2, where we mention that the batch-normalization weights and biases are kept dense. Dear reviewer, could you please elaborate your question more specifically so we can address your concerns? Thanks a lot!
>
>
> **Q3: Training epochs are large, which has negative societal impact and will limit its applications.**
>
> A3: Our proposed GaP method is to obtain the pruned model with the best quality. In our perspective, spending a bit more time in training to find a better model is well rewarded since models may be deployed to thousands of devices and inference billions of times.
> Based on the researches in Facebook [r3-1, r3-2], the models are inferenced many billions of times per day. Thus, a better pruning algorithm resulting to a smaller model may save a lot more computation on the inference side. As an example, the 90% non-uniform sparsity ResNet-50 model pruned using GaP achieves 77.9 top-1 accuracy on ImageNet, outperforming the previous SOTA of RigL-12x at 80% non-uniform sparsity by 0.5%. It also saves over 50% inference FLOPs with less training cost. Even if it is compared with the model that is trained for ~100 epochs at 80% sparsity (with 1.4% less accuracy), the extra training cost (900 epochs X 1M images per epoch) is only a tiny fraction of the inference cost in one day. The 50% inference FLOPs savings is a pure gain if the model is deployed for multiple days. Thus, we believe that our method advances the state-of-the-art in model compression efficiency.
>
> In our newly revised paper PDF, we have added a section in Appendix E to state the limitation and social impact.
>
> **(Please continue to read part 2)**

---

> > ### Comment · Reviewer_Zewc · 2021-11-16
> > **Response**
> >
> > Q2: In your network of experiment, they contain batch normalization layer. The masked features will lead to frequent fluctuations for the BN statistics. Did you have take this point into account in your methods and experiments?

---

> > > ### Author Response · Authors · 2021-11-16
> > > **Response to Reviewer Zewc**
> > >
> > > Dear reviewer,
> > >
> > > Thank you for your clarification. You are right that the pruning stage reduces the number of non-zero weights in the dense partition, and creates an abrupt change to the output activations, which in term affects the BN statistics calculation. We did not consider such fluctuations specifically in our experiments due to the following reasons.
> > >
> > > 1. The mean and variance are averaged among many minibatches, so the rate of change is slow. Since the partitions are sparse the majority of the training time, the mean and variance are closer to the sparse ones than the dense ones. Please note, when we grow a partition, the weights are grown from zeros, so that it does not affect the immediate statistics. In addition, only the weights having large impacts to model accuracy may grow to large magnitudes, and the rest are more likely to maintain small magnitudes so their impact to statistics may be low.
> > >
> > > 2. The GaP method only explores the sparse patterns. The BN statistics do not inference uniform sparsity, as the sparse pattern is purely determined by the weight values inside the same layer. It may only impact non-uniform sparsity, where the number of weights may flow from layer to layer. In this case, the BN statistics may indirectly affect the model accuracy via the learnable scaling and shifting factors. We consider it a secondary effect on model accuracy, as the abrupt change of weight values in the pruning stage affects the immediate model accuracy more.  To mitigate such impacts, we also fine-tune the model after the sparse pattern is fixed.
> > >
> > > 3. Experiments have shown that the model accuracy is superior to prior works without considering such statistics fluctuations. If we take such fluctuation into account, the accuracy may be even higher. We leave this to our future work.
> > >
> > > Thanks a lot for your question. We wonder whether we have addressed your concerns.

---

> ### Author Response · Authors · 2021-11-15
> **Response to Reviewer Zewc**
>
> ## Response - Part 2/2
>
> **(Continue from part 1)**
>
> **Q4: The practical implications of researching unstructured sparsity.**
>
> A4: In this paper, we focus on the unstructured sparsity to demonstrate the algorithm-level innovations of our proposed GaP method. We consider it an important line of work since it provides the theoretical limits for the structured sparsity pruning methods. More researches on practical structured sparsity may be built on top of the findings on the unstructured sparsity. Many other works from the recent top conferences such as ICML and NeurIPS [r3-3,r3-4,r3-5,r3-6 ] on pruning and sparse mask exploration methods fall into this category, and we use the same approach to compare with them.
>
> We recognize that the structured sparsity is a natural extension of our work, and we are currently working on that topic. We agree that the structured sparsity is beneficial to hardware acceleration, and many works such SCOP and HRank achieve good performance, though the methods used in those works are very different from ours. We have added those references in our related works section. Please note that we have evaluated the structured block sparsity with GaP method on Transformer and have achieved 27.2 SacreBLEU score (please refer to in Table 9 in the appendix). This type of structured sparsity is beneficial to the modern SIMD parallel processing architecture.
>
> [r3-1] Applied Machine Learning at Facebook: A Datacenter Infrastructure Perspective
>
> [r3-2] Machine Learning at Facebook: Understanding Inference at the Edge
>
> [r3-3] Liu, Shiwei, et al. "Do we actually need dense over-parameterization? in-time over-parameterization in sparse training." ICML 2021.
>
> [r3-4] Dettmers, Tim, and Luke Zettlemoyer. "Sparse networks from scratch: Faster training without losing performance." NeurIPS 2019.
>
> [r3-5] Tanaka, Hidenori, et al. "Pruning neural networks without any data by iteratively conserving synaptic flow." NeurIPS 2020.
>
> [r3-6] Sanh, Victor, Thomas Wolf, and Alexander M. Rush. "Movement pruning: Adaptive sparsity by fine-tuning." NeurIPS 2020.

---

> ### Author Response · Authors · 2021-11-21
> **Do you have more feedback?**
>
> Dear reviewer Zewc,
>
> Thanks a lot for spending time reviewing our paper and providing many insightful suggestions, including your follow up question. Now with the deadline of modifying the paper approaching, we wonder whether our responses have addressed your questions, and whether you have additional feedback. We'd like to clarify any of your confusion or concern before the paper discussion.
>
> Could you please let us know?
>
> Thanks a lot!

---

### Official Review · Reviewer_xnEk · 2021-11-02

**Correctness:** 3
**Technical Novelty And Significance:** 3
**Empirical Novelty And Significance:** 2
**Recommendation:** 6
**Confidence:** 3

**Main Review:**

Strengths:

+ The key contribution of this paper is the grow-and-prune schedule that does not need the dense pre-training initially. Such a design can help to reduce the time/computation cost during training.

+ Such a proposed design has ensured that all the weights could be updated during training, which is essential to the model convergence. Moreover, the authors provide the theoretical analysis, and it makes sense intuitively.

+ There are rich and solid experiments provided in this paper. It has been verified on both 2D/3D vision and NLP tasks which are quite strong to show the effectiveness of the proposed method. And there is a detailed ablation study in the appendix, which is also helpful.

+ This paper is written with clear logic and is easy to follow. The discussion of related works is comprehensive to include most of the recent and classic works.

Concerns/Questions:

1.	To verify the major claim in training efficiency, some of the important baselines are missing. It should be compared with the dense pre-trained model pruning in the time cost. For instance, the baseline’s total time should include pre-training time + pruning time + finetuning time. And please compare yours (pruning time + finetuning time) with it. Moreover, the performance of dense pre-trained model pruning is better provided for comparison.
2.	Since it has been claimed that the superiority of the proposed method comes from updating all the weights, I am not quite convinced about this point. As shown in Fig.2, it seems that the random mask exploration has not been limited within certain regions, which indicates that any parameters of the network can be updated. Although such a random mask exploration has not guaranteed that all the weights will be updated within some rounds, it looks like all of them will be updated in the end if there are enough epochs for exploration. Does this randomness destroy the convergence? Could you please explain more on this point?
3.	The partitions of parameters seem tricky in both partitions number and boundaries. Could you do more ablations on such two factors (and ResNet-50 may be a good choice for this ablation study)? And it will be better to share some high-level instructions on how to set such two hyper-parameters.
4.	What is the reason for the performance divergence between P-GaP and C-GaP? And how much time/computation costs have been saved comparing P-GaP with C-GaP.
5.	Ablation study on different sparsity (0%:10%:90%) may also be necessary. Any datasets are welcome.
6.	The last “s” of “previous works” of Remark 3 in Page 8 is red.
7.	Please make sure all the references are in the same format.

**Summary Of The Paper:**

This paper presents a new framework for model pruning. Unlike most previous pruning works, this framework does not require pre-training a dense model initially. They have proposed a schedule grow-and prune strategy to fulfill such a goal where the whole network has been split into multiple partitions. There will be an alternation of pruning and growing during the pruning, which ensures that all network parameters are updated in certain loops. Moreover, the authors provide theoretical insight behind such a design that all the weights trained per round are crucial for convergence. To accelerate the pruning process, this paper presents a parallel version that the growing and pruning of multiple partitions would be computed simultaneously. The accuracy and efficiency of the proposed framework have been verified on multiple tasks, including 2D images classification/detection, 3D detection, and text understanding.

**Summary Of The Review:**

I recommend borderline reject due to the missing of important baselines and I will consider changing my rating after seeing strong rebuttal.

---

> ### Author Response · Authors · 2021-11-15
> **Response to Reviewer xnEk**
>
> ## Response - Part 1/2
>
> Dear reviewer,
>
> Thank you very much for your effort reviewing our paper. Your valuable comments will significantly improve the clarity of our work. We have revised our paper PDF (changes are highlighted in blue).
>
> **Q1: To claim training efficiency, the results need to be compared with the dense pre-trained model pruning in time cost.**
>
> A1: Dear reviewer, in Table 1, we have listed the training epochs of our work and all related works on the ResNet-50 model. Specifically, the prune-from-dense training methods cost 750 epochs for 80% sparsity, which contains 250 epochs for pretraining the dense model, 250 epochs of pruning using ADMM, and 250 epochs of fine-tuning. The 90% sparsity model is progressively pruned from the fine-tuned 80% sparsity model, with a total training cost of 1250 epochs. The detailed training recipe is described in Appendix B.1. Similarly, the prune-from-dense training cost of SSD on COCO is discussed in Appendix B.3. In our comparisons, we list the accuracy of the dense models in Tables 1-4. If you’d like to see the training cost for the dense models, we have described them in more detail to Appendix B. If you’d like to see more comparisons, please let us know.
>
> Please note, we do not claim that our method saves training time. Instead, we try to achieve the best possible sparse model quality. Based on the researches in Facebook [r2-1,r2-2], the models are inferenced many billions of times per day. Thus, a better pruning algorithm resulting to a smaller model may save a lot more computation on the inference side. As an example, the 90% non-uniform sparsity ResNet-50 model pruned using GaP achieves 77.9 top-1 accuracy on ImageNet, outperforming the previous SOTA of RigL-12x at 80% non-uniform sparsity by 0.5%. It also saves over 50% inference FLOPs with less training cost. Even if it is compared with the model that is trained for ~100 epochs at 80% sparsity (with 1.4% less accuracy), the extra training cost (900 epochs X 1M images per epoch) is only a tiny fraction of the inference cost in one day. The 50% inference FLOPs savings is a pure gain if the model is deployed for multiple days. Thus, we believe that our method advances the state-of-the-art in model compression efficiency.
>
>
>
> **Q2: Enough epochs might guarantee full exploration.**
>
> A2: In our scheduled GaP algorithm, we intend to use all the information in the model. i.e., explore all weights efficiently. It is true that when trained sufficiently long, all weights will be explored. However, such training cost may be prohibitive when explored randomly. Random exploration and our scheduled GaP method are like sampling operations with replacement or without replacement, and they are greatly different.
>
> For example, if we consider a very small model with only 10 weights, and each time we only train one weight and keep the remaining weights the same. When we apply the scheduled GaP algorithm (without replacement), each weight is guaranteed to be trained after 10 steps (one step indicates training one weight). This is because the scheduled GaP alternates the weight to train. The already trained weight is guaranteed not to be re-trained again until all the remaining weights are trained. On the other hand, if a random weight is selected to train in each step (with replacement), it may take 29 steps to be fairly confident that all weights are trained. That is almost 3 times more training time than the scheduled GaP. Please note, in our example, we only have 10 weights. A real neural network contains millions of weights and it is more difficult to guarantee a full exploration.
>
> Thus, we believe that our method explores the weight more efficiently than the random exploration methods. Thanks for your question. We have added this discussion in Appendix C the revised paper PDF.
>
>
>
> **Q3: Please evaluate the partition number and boundaries more.**
>
> A3: We have evaluated the different partition numbers in Appendix A.1. For ResNet-50 model, partitions 1 and 4 are analyzed (Table 5). For the Transformer model, partitions 1, 3, and 6 are analyzed (Table 6). In both scenarios, we find that one partition is not sufficient to get good results. This shows that we cannot rely on DSD[r2-3] like mechanisms alone. Instead, the model needs to be partitions. On the other hand, in the Transformer model, we find that the 3-partition model achieves the best SacreBLEU score, and the 6-partition model results to worse quality. It is an indication that the best number of partitions is bounded. We didn’t perform experiments with more partitions since it takes longer to train with more partitions. We find that a few partitions are good enough to get good performance. Empirically, a small number of equal partitions that naturally follow the structures of the models usually perform well.
> **(Please continue to read part 2)**

---

> ### Author Response · Authors · 2021-11-15
> **Response to Reviewer xnEk**
>
> ## Response - Part 2/2
>
> **(Continue from part 1)** For the partition boundaries, we try to equally partition the model with similar parameter counts or computation, and we also intuitively group the consecutive layers as the partitions. We hypothesize that the adjacent layers are tighter correlated than the distant layers. For example, the weight W_key, W_query, W_value in the same attention block in a Transformer are strongly correlated since they process the same input sequence for the self-attention computation. We compare the performance of Transformer using the consecutive grouping partitions (i.e., the C-GaP or P-GaP in our paper) and random partitions. We find that the model with the consecutive grouping partition method achieves 27.7 SacreBLEUscore in uniform 0.9 sparsity ratio, while the model with random partition only achieves 27.0 and 26.9 SacreBLEUscore with uniform and non-uniform 0.9 sparsity ratio, respectively.
>
> Dear reviewer, we hope that our added analysis better explains the effect of the partition number and partition boundaries. Please let us know if you have better suggestions. Thanks a lot!
>
>
> **Q4: P-GaP and C-GaP difference in accuracy, training cost and training time.**
>
> A4: P-GaP convergences slower than C-GaP. The reason is that in C-GaP when one dense partition is explored, the weights in the other sparse partitions are still trained to get better model accuracy. On the other hand, the P-GaP method only keeps the dense partition and discards the sparse partitions when combining all dense partitions to the updated  model. In our experiments, we use same number of training epochs for P-GaP and C-GaP, which is an advantage to C-GaP. Even though the number of training epochs for P-GaP and C-GaP are the same, the training time for P-GaP is much less, because P-GaP explores all partitions parallelly in different machines, while C-GaP uses a cyclic traversal paradigm that explores partitions one by one sequentially. That is, P-GaP trades more hardware resources for shorter training time. Thanks for your clarification request.
>
>
> **Q5: More ablation on different sparsity.**
>
> A5: In our experiments, we choose 80% and 90% sparsity in order to compare with other works since they are the mainstream sparsity evaluated in many piror pruning methods (e.g., GMP, SNIP, GraSP, DeepR, SET, RigL, DSR in our paper, and other works [r2-4, r2-5, r2-6, r2-7]). Based on our experience, when the sparsity ratio is less than approximately 60%, the accuracy is similar with the dense network and the choice of pruning methods matters less. For example, NVIDIA Ampere GPU architecture supports a special 2:4 sparse pattern (50% sparsity ratio) [r2-8] which results to similar accuracies with the dense model using a simple one-shot magnitude pruning method. Additionally, there are hardly any acceleration gains if the sparsity ratio is low. In other words, the effect of different pruning methodologies can only be observed when the sparsity is high, as some pruning methods result to accuracy loss. Based on the above reasons, we select to use 80% and 90% sparsity ratio in our experiments. We agree with you that more data are better, but with limited computing resources and time, it would be difficult to add complete the sparsity sweep experiments in time and draw a definite conclusion. Thus, we plan to complete the experiments in the future.
>
>
> **Q6/7: Minor errors and reference format.**
>
> A6/7: Thank you for pointing them out. We have corrected the minor errors and reference format you mentioned directly in our revised paper PDF.
>
>
> [r2-1] Applied Machine Learning at Facebook: A Datacenter Infrastructure Perspective
>
> [r2-2] Machine Learning at Facebook: Understanding Inference at the Edge
>
> [r2-3] Han, Song, et al. "DSD: Dense-sparse-dense training for deep neural networks." ICLR 2017.
>
> [r2-4] Liu, Shiwei, et al. "Do we actually need dense over-parameterization? in-time over-parameterization in sparse training." ICML 2021.
>
> [r2-5] Dettmers, Tim, and Luke Zettlemoyer. "Sparse networks from scratch: Faster training without losing performance." arXiv preprint arXiv:1907.04840 (2019).
>
> [r2-6] Tanaka, Hidenori, et al. "Pruning neural networks without any data by iteratively conserving synaptic flow." NeurIPS 2020.
>
> [r2-7] Sanh, Victor, Thomas Wolf, and Alexander M. Rush. "Movement pruning: Adaptive sparsity by fine-tuning." NeurIPS 2020.
>
> [r2-8] Mishra, Asit, et al. "Accelerating sparse deep neural networks." arXiv preprint arXiv:2104.08378 (2021).

---

> ### Author Response · Authors · 2021-11-20
> **Please advice on our response to your review**
>
> Dear reviewer xnEk,
>
> Thanks a lot for spending time reviewing our paper and providing many insightful suggestions. We have posted our responses and updated the paper several days ago. Now with the deadline of modifying the paper approaching, we wonder whether you have additional comments or questions on our responses. We'd like to clarify any of your confusion or concern before the paper discussion.
>
> Could you please let us know?
>
> Thanks a lot!

---

> > ### Comment · Reviewer_xnEk · 2021-11-27
> > **Response to Rebuttal**
> >
> > Dear Authors,
> >
> > Thanks for your hard work on the rebuttal. I still have two minor issues. First of all, it seems that the FLOPs are highly related to the sparsity ratio (see Table 1), which makes me confused. While in the unstructured pruning, as claimed by Reviewer Zewc, the FLOPs should have minor relation with the sparsity. Could you please share more details about how to calculate the FLOPs in this paper? Another concern is that there is a kind of over-claim in the revised content, e.g., "It targets to obtain the best quality sparse model under a runtime budget." and "The scheduled GaP methods target to obtain the best quality of the pruned model under a runtime budget." Although the proposed model shows superiority over the baselines (mostly RigL), it is hard to say it is the "best quality" since there is still a lot to explore in this area.

---

> > > ### Author Response · Authors · 2021-11-27
> > > **Response to reviewer xnEk**
> > >
> > > Dear reviewer xnEk,
> > >
> > > Thank you very much for your additional questions. Below are our responses:
> > >
> > > **Q1: Explanation on the relations of FLOPs and sparsity**
> > >
> > > **A1**: We could not find reviewer Zewc's question on the relations of FLOPs and sparsity. Instead, we found: "However, the FLOPs has minor relations with **inference speed**". It questions the practical implications of unstructured sparsity. We fully agree with this comment. Please refer to our response to Zewc's question for a detailed explanation.
> > >
> > > We understand that unstructured sparsity has irregular weight distribution that is not friendly to practical hardware acceleration. Thus the calculated FLOPs cannot fully reflect the inference speed at hardware-level. However, in most of the pruning works on unstructured sparsity, FLOPs is a key indicator to demonstrate the algorithm-level innovation. Because it provides the theoretical limits for the structured sparsity pruning methods that are practical for hardware acceleration.
> > >
> > > In order to calculate the theoretical FLOPs of a pruned model, we count the operations involved with the non-zero weights. For example, if a DNN layer is 80% pruned, we count the total number of operations from the 20% non-zero weights. We compute the FLOPs layer-wisely and add them up to get the overall FLOPs of a pruned model.
> > >
> > > Please note that FLOPs is not one-to-one mapped to sparsity. Different non-zero weights may contribute to different FLOPs (i.e. their reuse factors are different). Also, different layers may contain different number of non-zero weights. These two factors may cancel out on some models (e.g. ResNet50). Different layers have different number of non-zero weights, each with a different reuse factor. However, their total contribution to FLOPs is about the same across layers. Thus, it gives the perception that FLOPs and sparsity are linearly correlated. This is a special case for ResNet50. In some other models, such correlation may not be linear. Nevertheless, in our opinion, FLOPs and sparsity are highly correlated.
> > >
> > > We have explained the relations of FLOPs and sparsity based on our understanding. However, we are still confused on your claim that FLOPs have minor relation with sparsity. We wonder whether you can further clarify your concerns so we can address them directly. Thanks.
> > >
> > >
> > > **Q2: Overclaim on quality**
> > >
> > > **A2**: We are sincerely sorry about the misunderstanding on overclaiming. We did not intend to claim that our method was the best. We fully agree with you that there are still a lot of room for improvement. Our initial intention was to clarify that we tried to get a better model without training resource constraints. Thus, we will revise the wording to: "The goal of the paper is to reduce the accuracy loss of the pruned models, given sufficient training resources". We hope this will relieve your concerns.
> > >
> > >
> > > Dear reviewer xnEk, we genuinely hope our responses answer your questions. If you need further clarification, please feel free to comment back. Again, we are very thankful for your time and all the constructive feedback!

---

> > > > ### Comment · Reviewer_xnEk · 2021-11-29
> > > > **Further Response**
> > > >
> > > > Dear Authors,
> > > >
> > > > Thanks for your reply. Most of my concerns are addressed through rebuttal except the FLOPs, which is not a strong standpoint. But considering the overall contributions, I decided to change the score to 6.

---

> > > > > ### Author Response · Authors · 2021-11-29
> > > > > **Author response to reviewer xnEk**
> > > > >
> > > > > Dear reviewer xnEk,
> > > > >
> > > > > We want to thank you for your appreciation of our work and for raising the score!  Your comments are very constructive, e.g., the discussion on methods that guarantee full exploration, partition number and boundaries, etc. We will make sure that all the discussion points are carefully addressed and included in our final manuscripts. The quality of our paper will be improved with your help.
> > > > >
> > > > > Thank you again for your valuable time!
> > > > >
> > > > > Best
> > > > > Authors

---

### Official Review · Reviewer_XG8a · 2021-11-02

**Correctness:** 4
**Technical Novelty And Significance:** 2
**Empirical Novelty And Significance:** 2
**Recommendation:** 6
**Confidence:** 4

**Details Of Ethics Concerns:**

None.

**Main Review:**

Strengthes:
1. The empirical results demonstrate the intuition that exploring all the weights could produce a better pruning result compared to a random selection of weights that an algorithm would explore. I am curious when a random selection algorithm is used, if we apply heuristics such as recording the selected weight to make sure all the weights are explored, would that lead to a similar result?
2. A key variable of the proposed approach is the number of partition. Some empirical studies are done and presented in the supplementary results and a four-partition GaP is suggested. I wonder when the network size is big, whether the grow-and-prune process is slow within each partition?

Weakness:
1. The approach has several key variables and those variables are likely to impact the results. This makes it not easy for later studies to build upon this work.
2. As presented in Supplementary results A2, Table 8, it seems that partition strategy doesn’t matter much. Does that mean any approach that could guarantee a complete transverse of the weights, it would produce a better pruning result?


**Summary Of The Paper:**

This paper presents a scheduled grow-and-prune approach to produce a sparse CNN model. In particular, a sparse model is achieved by repeatedly growing and prune subsets of layers during network training. The main difference between this approach and other approaches that do pruning during the training time is that this approach guarantees that all the weights are explored when a pruning decision is made.

In the paper, two variations of the approaches are discussed: C-GaP and P-GaP. The theoretical convergence of the C-GaP is analyzed.
Experimental results on various datasets and tasks are presented, which shows improvement compared to baseline approaches. Additionally, key variables such as partition number are explored and empirical results are provided in the supplementary results.

**Summary Of The Review:**

Empirical results and conclusions that are presented in this paper are interesting. Even if some of the theoretical analyses were made, the experiments are mostly empirical. Therefore, I believe the paper is useful for researchers in this area to read, but this approach doesn't make a breakthrough. Therefore, I rate it 6.

---

> ### Author Response · Authors · 2021-11-15
> **Response to Reviewer XG8a**
>
> Dear reviewer,
>
> Thank you for spending time reviewing our paper and providing many valuable suggestions. We have revised our paper PDF (changes are highlighted in blue). Below are our summarized questions and responses.
>
> **Q1: What if heuristics are used to guarantee all weights are explored? Random partition doesn't seem to be worse.**
>
> A1: In the scheduled GaP algorithm, the interface between the dense partition and the sparse partition forms a discontinuity boundary. When the dense partition is pruned to sparse, it creates an abrupt change to the model and negatively affects the model quality. Thus, some amount of training is required to recover from it. Intuitively, it is better to reduce such discontinuity boundary to preserve accuracy. The extent that this discontinuity affects the model quality is model specific. We have performed an ablation study in Table 7 showing that for ResNet-50, the random partition is slightly worse than the scheduled cyclic partition (top-1 accuracy is 77.8% vs 77.9%). This small difference may be due to the characteristics of ResNet-50. After the paper deadline, we have also explored random partition on the Transformer model, and the accuracy gap is much larger. At 90% sparsity, Transformer model can achieve 27.7 SacreBLEU scores in uniform sparsity with 3 partitions using C-GaP (Table 4). When a random 3-partition is applied to a uniform and non-uniform 90% sparsity Transformer, the SacreBLEU score is reduced to 27.0 and 26.9, respectively. Those are -0.7 and  -0.8 SacreBLEU differences for uniform and non-uniform sparsity. We conjecture that this accuracy divergence is due to the inter-layer weight correlation is stronger in Transformer architecture (e.g., the W_key, W_query, W_value in the same attention block have very strong correlation) than in the ResNet architecture.
>
> The aforementioned comparison may also shed some light on the quality for random sparsities, where only a subset of the weights in a layer may be selected and trained at the same time. It creates more discontinuity boundaries, as the random layer partition is a subset. Even if we ensure that all weights are equally explored, it may converge to a stable point with worse accuracy.
>
> Thanks for your question. We have added the study on Transformers in the ablation section (Table 8) in our revised paper PDF.
>
>
>
> **Q2: When the network size is big, does the grow-and-prune process is slow.**
>
> A2: We are confused about this question. Please correct us if we do not address your question. From our experience, the grow-and-prune process only takes a few seconds since the partition growing stage only sets the corresponding mask to all ones, and the pruning stage sets the weights with the least magnitude to zero. We also do not observe noticeable convergence rate difference between large and small models, though training a small model is significantly faster than a large model. Again, we may have entirely missed your point. Feel free to comment back and we are looking forward to address your concerns.
>
>
> **Q3: Key variables have impact on results, hard for later studies to build upon it.**
>
> A3: Thanks for raising this question. It is indeed a very general and difficult problem in the field. Many works introduce hyper-parameters and rely them to achieve good results [r1-1, r1-2, r1-3].  For example, the weight update frequency and weight update ratio in RigL [r1-1] are empirically set. In our work, we have done the ablations on key hyper-parameters too, such as the weight partition strategy (group of consecutive layers or random partition), partition numbers, training schedule (cyclic or parallel), etc. With limited time and compute resources, our evaluation could not thoroughly cover all the hyper-parameter settings, so we explored the most relevant ones based on our experience. In our future work, we will focus on how to tune those parameters and evaluate their impact to the network performance using a more systematic approach, such as AutoML. Dear reviewer, again thank you very much for your time. Please feel free to ask if you have more questions.
>
>
> [r1-1] Evci, Utku, et al. "Rigging the lottery: Making all tickets winners." ICML 2020.
>
> [r1-2] Frankle, Jonathan, and Michael Carbin. "The lottery ticket hypothesis: Finding sparse, trainable neural networks." ICLR 2019.
>
> [r1-3] Zhang, Tianyun, et al. "A systematic DNN weight pruning framework using alternating direction method of multipliers." ECCV 2018.

---

> ### Author Response · Authors · 2021-11-20
> **Please advice on our responses to your review**
>
> Dear reviewer XG8a,
>
> Thanks a lot for spending time reviewing our paper and providing many insightful suggestions. We have posted our responses and updated the paper several days ago. Now with the deadline of modifying the paper approaching, we wonder whether you have additional comments or questions on our responses. We'd like to clarify any of your confusion or concern before the paper discussion.
>
> Could you please let us know?

---

### Decision · Program_Chairs · 2022-01-20

**Decision:**

Accept (Poster)

**Comment:**

The paper proposes a methodology for alternatively growing and pruning a subset of layers within a network in order to eventually produce a trained, sparse model.  After discussion, all reviewers favor accept.  Empirical performance of the sparse models appears strong, but requires significant computational expense during training to achieve.